# The role of cochlear place coding in the perception of frequency modulation

Kelly L Whiteford*, Heather A Kreft, Andrew J Oxenham

Department of Psychology, University of Minnesota, Minneapolis, United States

**Abstract** Natural sounds convey information via frequency and amplitude modulations (FM and AM). Humans are acutely sensitive to the slow rates of FM that are crucial for speech and music. This sensitivity has long been thought to rely on precise stimulus-driven auditory-nerve spike timing (time code), whereas a coarser code, based on variations in the cochlear place of stimulation (place code), represents faster FM rates. We tested this theory in listeners with normal and impaired hearing, spanning a wide range of place-coding fidelity. Contrary to predictions, sensitivity to both slow and fast FM correlated with place-coding fidelity. We also used incoherent AM on two carriers to simulate place coding of FM and observed poorer sensitivity at high carrier frequencies and fast rates, two properties of FM detection previously ascribed to the limits of time coding. The results suggest a unitary place-based neural code for FM across all rates and carrier frequencies.

## Introduction

Modulations in frequency (FM) and amplitude (AM) carry critical information in biologically relevant sounds, such as speech, music, and animal vocalizations (*Attias and Schreiner, 1997*; *Nelken et al., 1999*). In humans, AM is crucial for understanding speech in quiet (*Shannon et al., 1995*; *Smith et al., 2002*), while FM is particularly important for perceiving melodies, recognizing talkers, determining speech prosody and emotion, and may aid in the perception of speech presented in competing background sounds (*Zeng et al., 2005*; *Strelcyk and Dau, 2009*; *Sheft et al., 2012*; *Johannesen et al., 2016*; *Lopez-Poveda et al., 2017*; *Parthasarathy et al., 2019*). The perception of FM at both slow and fast modulation rates is often degraded in older people and those with hearing loss (*Lacher-Fougère and Demany, 1998*; *Moore and Skrodzka, 2002*; *He et al., 2007*; *Strelcyk and Dau, 2009*; *Grose and Mamo, 2012*; *Paraouty et al., 2016*; *Wallaert et al., 2016*; *Paraouty and Lorenzi, 2017*; *Whiteford et al., 2017*). This deficit likely contributes to the communication difficulties experienced by such listeners in noisy real-world environments, which may in turn help explain why age-related hearing loss has been associated with decreased social engagement, greater rates of cognitive decline, and increased risk of dementia (*Lin et al., 2011*; *Lin et al., 2013*; *Lin and Albert, 2014*; *Deal et al., 2017*; *Thomson et al., 2017*). Current assistive listening devices, such as hearing aids and cochlear implants, have been generally unsuccessful in restoring FM sensitivity (*Chen and Zeng, 2004*; *Ives et al., 2013*). This lack of success may be related to a gap in our scientific understanding regarding how FM is extracted by the brain from the information available in the auditory periphery.

The coding of AM begins in the auditory nerve with periodic increases and decreases in the instantaneous firing rate of auditory nerve fibers that mirror the fluctuations in the temporal envelope of the stimulus (*Schreiner and Langner, 1988*; *Joris et al., 2004*). As early as the inferior colliculus and extending to the auditory cortex, rapid AM rates are transformed to a code that is no longer time-locked to the stimulus envelope and instead relies on overall firing rate, with different neurons displaying lowpass, highpass, or bandpass responses to different AM rates (*Schreiner and Langner, 1988*; *Nelson and Carney, 2007*; *Wang et al., 2008*). The coding of FM is less straightforward. For a pure tone with FM, the temporal envelope of the stimulus is flat; however, the changes

*For correspondence:
whit1945@umn.edu

Competing interests: The authors declare that no competing interests exist.

in frequency lead to dynamic shifts in the tone's tonotopic representation along the basilar membrane, resulting in a transformation of FM into AM at the level of the auditory nerve (*Zwicker, 1956*; *Khanna and Teich, 1989*; *Moore and Sek, 1995*; *Saberi and Hafter, 1995*; *Sek and Moore, 1995*).

Although this FM-to-AM conversion provides a unified and neurally efficient code for both AM and FM (*Saberi and Hafter, 1995*), it falls short of explaining human behavioral trends in FM sensitivity. Specifically, at low carrier frequencies ($f_c$ <~4–5 kHz) and slow modulation rates ($f_m$ <~10 Hz) FM sensitivity is considerably greater than at higher carrier frequencies or fast modulation rates in a way that is not predicted by a simple FM-to-AM conversion mechanism (*Demany and Semal, 1989*; *Moore and Sek, 1995*; *Moore and Sek, 1996*; *Sek and Moore, 1995*; *He et al., 2007*; *Whiteford and Oxenham, 2015*; *Whiteford et al., 2017*). This discrepancy is important, because it is FM at low frequencies and slow modulation rates that is most critical for human communication, including speech and music, as well as many animal vocalizations (*Attias and Schreiner, 1997*; *Nelken et al., 1999*). The enhanced sensitivity to slow FM at low carrier frequencies has been explained in terms of an additional neural code based on stimulus-driven spike timing in the auditory nerve that is phase-locked to the temporal fine structure of the stimulus (*Rose et al., 1967*; *Moore and Sek, 1995*; *Parthasarathy et al., 2019*). Although a code based on time intervals between phase-locked neural spikes can potentially provide greater accuracy (*Siebert, 1970*; *Heinz et al., 2001*), be maintained to some extent in the auditory brainstem (*Paraouty et al., 2018*), and be used for spatial localization (*Moiseff and Konishi, 1981*; *Grothe et al., 2010*), it is not known whether or how this timing information is extracted by higher stages of the auditory system to encode periodicity (pitch) and FM.

If the detection of FM at fast rates depends on an FM-to-AM conversion, whereas the detection of FM at slow rates does not, then fast-rate FM detection thresholds should depend on the sharpness of cochlear tuning (*Figure 1*), whereas slow-rate FM detection thresholds should not. Previous studies using normal-hearing listeners have not demonstrated such a relationship for either slow or fast FM rates (*Whiteford and Oxenham, 2015*; *Whiteford et al., 2017*). However, this failure to find a correlation may be due to a lack of variability in cochlear filtering within the normal-hearing population. *Johannesen et al., 2016* found only a modest correlation between slow-rate FM ($f_c$ = 1500 Hz; $f_m$ = 2 Hz) and cochlear mechanical gain loss, but FM thresholds were measured in the presence of superimposed AM, which could have increased the between-subject variability in the measurements (e.g., *King et al., 2019*). Furthermore, FM sensitivity was not measured at a faster rate, where only place cues are thought to be utilized. People with cochlear hearing loss often have poorer frequency selectivity (*Glasberg and Moore, 1986*; *Moore et al., 1999*), due to a broadening of cochlear tuning (*Robertson and Manley, 1974*; *Liberman et al., 1986*; *Moore, 2007*). By contrast, damage to the cochlea is not thought to lead to a degradation of auditory-nerve phase locking to temporal fine structure for sounds presented in quiet (*Henry and Heinz, 2012*; *Henry et al., 2019*), so we would not expect to find a strong relationship between slow-rate FM detection thresholds and hearing-loss-induced changes in cochlear tuning if slow-rate FM relies primarily on a time code.

Experiment 1 measured FM and AM detection thresholds at slow ($f_m$ = 1 Hz) and fast ($f_m$ = 20 Hz) modulation rates in a large sample of listeners with hearing thresholds at the carrier frequency ($f_c$ = 1 kHz) ranging from normal (~0 dB sound pressure level, SPL) to severely impaired (~70 dB SPL), consistent with sensorineural hearing loss (SNHL). The fidelity of cochlear frequency tuning was assessed using a psychophysical method to estimate the slopes of the forward masking pattern around 1 kHz (e.g., *Kidd and Feth, 1981*). The results revealed a relationship between the estimated sharpness of cochlear tuning and sensitivity to FM at both fast and slow modulation rates, suggesting that place coding fidelity directly affects FM sensitivity. This relationship remained significant even after controlling for potentially confounding factors such as degree of hearing loss, sensitivity to AM, and age.

Experiment 2 provided a direct test of earlier assumptions that had led to the conclusion that phase-locked timing information is necessary to code slow FM. We simulated important aspects of the cochlear response to FM without the presence of auditory-nerve timing cues by presenting two tones, spaced far enough away from each other in frequency to avoid peripheral interactions, and applying out-of-phase AM to them. This resulted in a simulation of the out-of-phase AM produced by a single FM tone. Sensitivity to the out-of-phase AM mirrored that seen in traditional psychophysical studies of FM detection, with sensitivity highest at low frequencies and slow rates, despite the

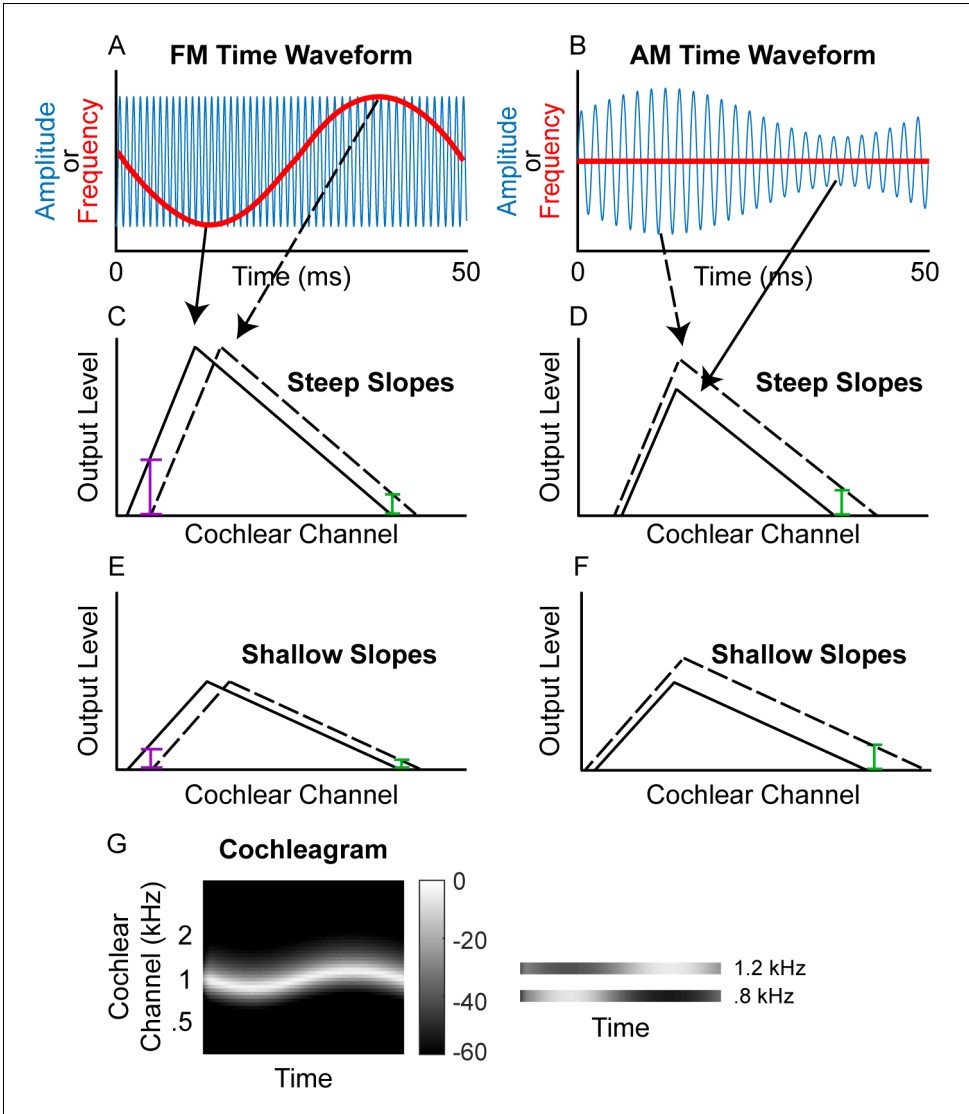

**Figure 1.** Schematic of (A) FM and (B) AM time waveforms ($f_c$ = 1 kHz; $f_m$ = 20 Hz) and the resulting changes in basilar-membrane excitation for steep (C and D) and shallow (E and F) slopes. In A and B, the blue time waveforms represent amplitude over time, while the superimposed red waveforms are the same stimuli plotted in terms of instantaneous frequency over time. Panels C and E demonstrate that a place code for FM would result in a greater change in output level on the low-frequency side of the excitation pattern (purple bars) relative to the high-frequency side (green bars) and that shallower filter slopes result in poorer FM coding (larger colored bars in C than in E) but not poorer AM coding (same size colored bars in D and F). (G) Schematic cochleagram of an FM tone, showing how the output from two separate cochlear channels (right) with center frequencies on either side of the carrier frequency is AM that is out of phase.

lack of any usable timing cues based on temporal fine structure. Taken together, our results suggest that a time-interval code is not necessary to represent slow-rate FM and instead imply a unitary neural code for FM across all rates and frequencies.

## Results

### Experiment 1
Relationship between hearing loss and frequency selectivity

The fidelity of place coding at the test frequency (1 kHz) was measured using pure-tone forward-masking patterns. The 56 participants had to detect a brief tone pip that followed a masker tone presented at 1 kHz. The level of the tone pip was adapted to track its detection threshold. Without the presence of a masker, the threshold level of the tone pip reflects the absolute threshold (*Figure 2—figure supplement 1*, unfilled circles). In the presence of a pure-tone forward masker, the level of the tone pip depends on the tone pip's frequency proximity to the masker and on an individual's frequency selectivity, as determined by their cochlear tuning (*Shera et al., 2002*; *Sumner et al., 2018*). For each participant, the steepness of the low- and high-frequency slopes of the masking function was estimated via linear regression of the thresholds (in dB SPL) for the four tone-pip frequencies below (800, 860, 920, and 980 Hz) and above (1020, 1080, 1140, and 1200 Hz) the masker frequency. Within-subject test-retest reliability of the estimated slope functions was high (bootstrapped simulated test-retest correlations of $r = 0.98$ and $r = 0.953$ for the low and high slopes, respectively; see Materials and methods). The range of measured masking function slopes in the present study spanned 152 dB/octave for the low slope ($-24$ to 128 dB/octave; $\bar{x}=49.4$) and 120 dB/octave for the high slope ($-92.7$ to 28.3 dB/octave; *Figure 2*, y-axis; $\bar{x}=-23.3$), which was greater than the typical range in just normal-hearing listeners at 500 Hz (e.g., 128 dB/octave range for the low slope and 89 dB/octave range for the high slope in *Whiteford et al., 2017*).

Consistent with expectations (*Glasberg and Moore, 1986*), the amount of hearing loss at the tone-pip frequency correlated with the slopes of the masking functions (*Figure 2*; low slope: $r = -0.685$, p<0.0001, CI = $[-0.804, -0.513]$; high slope: $r = 0.717$, p<0.0001, CI = $[.559, .826]$), supporting the idea that hearing loss is associated with poorer frequency tuning. However, frequency tuning is believed to be governed solely by basilar membrane mechanics and outer hair cell function (*Ruggero and Rich, 1991*; *Sumner et al., 2018*), whereas overall hearing loss may include contributions from other factors, such as conductive loss or the function of the inner hair cells and the

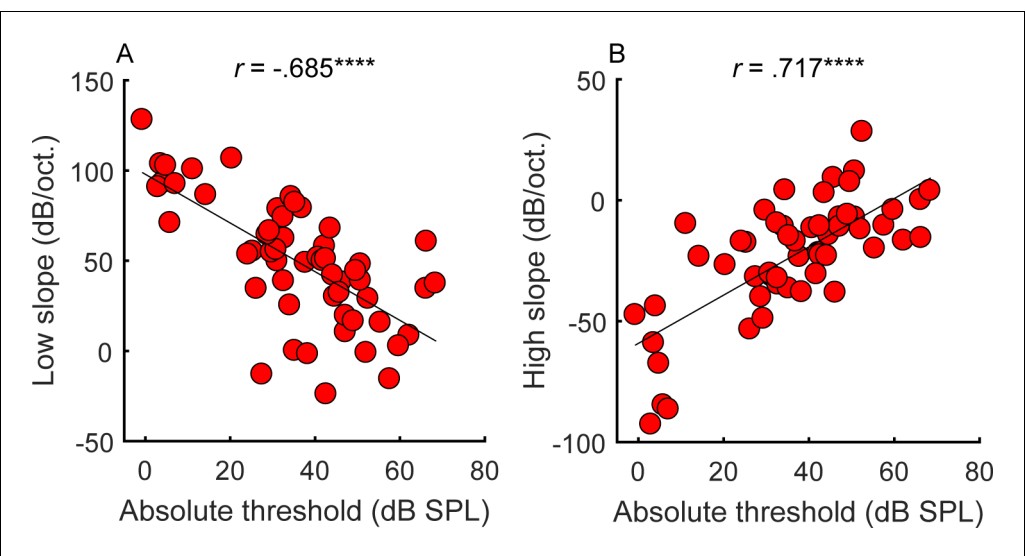

**Figure 2.** Correlations between average absolute thresholds at 1 kHz in the tested ear (x-axis) and the steepness of the (**A**) low and (**B**) high side of the cochlear filter slopes (n = 55). Participants with greater hearing loss at 1 kHz tended to have shallower filter slopes. Correlations marked with an * are significant after Holm's correction (****p<0.0001).

The online version of this article includes the following source data and figure supplement(s) for figure 2:

**Source data 1.** *Figure 2* data.

**Figure supplement 1.** Example forward masking pattern for a single subject.

**Figure supplement 1—source data 1.** Individual masking pattern.

auditory nerve. These additional factors may explain why filter slopes account for only approximately half the variance observed in absolute thresholds.

## Relationship between FM and AM detection

When compared to earlier results from normal-hearing listeners varying in age (*Whiteford et al., 2017*), the range of FM detection thresholds was much wider in the present study, whereas the range of AM detection thresholds was comparable (Figure 2 from *Whiteford et al., 2017* with the current *Figure 3*). This result confirms that hearing loss affects the detection of FM more than AM (*Lacher-Fougère and Demany, 1998*). Test-retest reliability for the estimation of AM and FM detection thresholds was very high (average correlations using a bootstrapping procedure: slow FM, $r = 0.973$, p<0.0001, CI = [.954, .984]; fast FM, $r = 0.97$, p<0.0001, CI = [.949, .983]; slow AM, $r = 0.925$, p<0.0001, CI = [.874, .956]; fast AM, $r = 0.956$, p<0.0001, CI = [.925, .974]). If slow FM

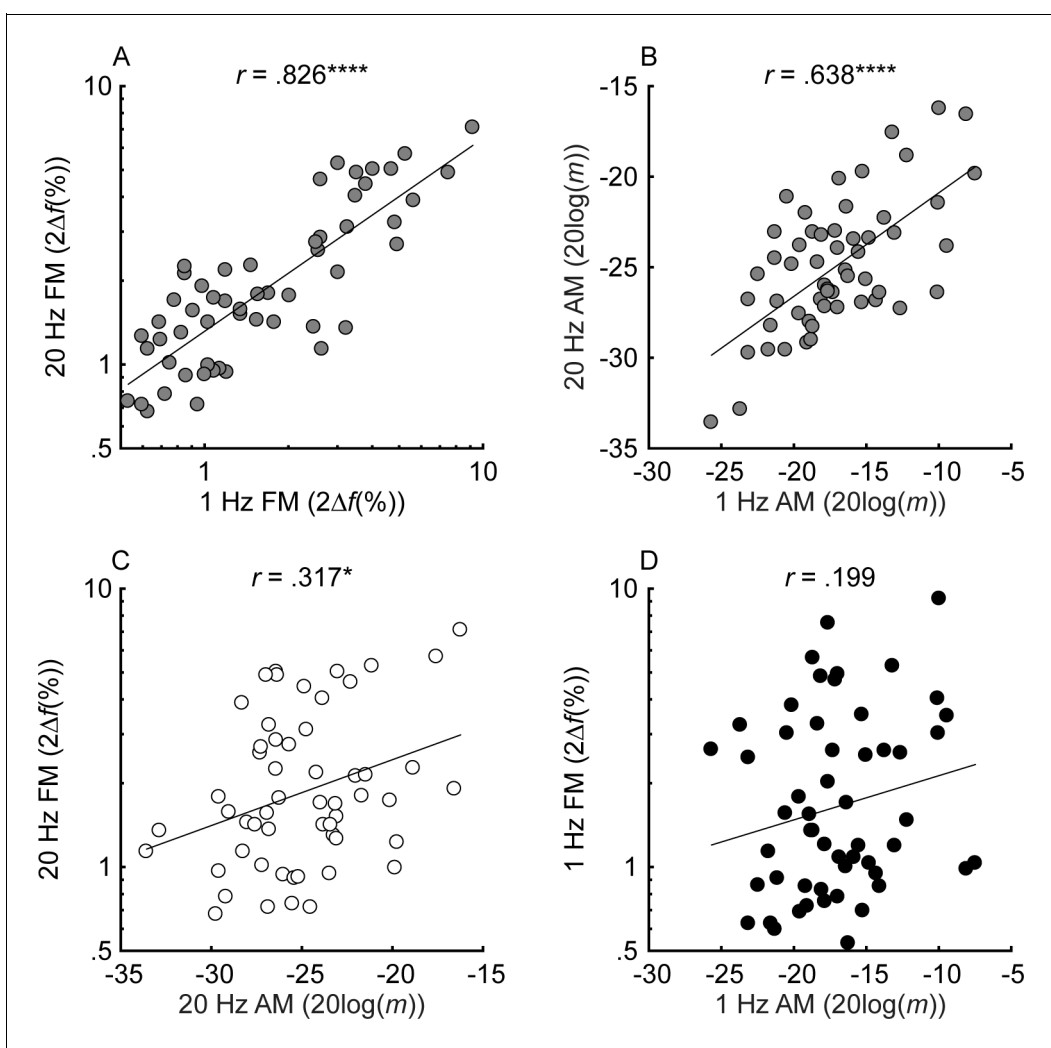

**Figure 3.** Individual thresholds for slow ($f_m$ = 1 Hz) and fast ($f_m$ = 20 Hz) FM and AM detection (n = 55). Black, white, and gray circles represent slow, fast, and mixed modulation rates, respectively. FM and AM thresholds are plotted in percent peak-to-peak frequency change ($2\Delta f(\%)$) and $20\log(m)$, where $\Delta f$ is the frequency excursion from the carrier and $m$ is the modulation depth (ranging from 0 to 1). For all tasks, lower values represent better thresholds. Shown in the different panels are the relationships between thresholds in slow and fast FM (**A**), slow and fast AM (**B**), fast AM and FM (**C**), and slow AM and FM (**D**). Correlations marked with an * are significant after Holm's correction (****p<0.0001, ***p<0.001, **p<0.01, and *p<0.05).

The online version of this article includes the following source data for figure 3:

**Source data 1.** *Figure 3* data.

utilizes a time-interval code, then across-listener variability in slow FM detection should partly reflect variability in time coding. This means that across-listener correlations in tasks thought to use a shared code (fast FM, slow AM, and fast AM) should be greater than in tasks thought to use different codes (slow FM with any other task). Inconsistent with this prediction, slow and fast FM detection thresholds were strongly correlated ($r = 0.826$, p<0.0001, CI = [.718, .895]; *Figure 3*, panel A). Significant correlations were also observed between detection thresholds for slow and fast AM ($r = 0.638$, p<0.0001, CI = [.449, .773]; *Figure 3*, panel B) fast FM and fast AM ($r = 0.317$, p=0.028, CI = [.056, .537]; *Figure 3*, panel C) and fast FM and slow AM ($r = 0.366$, p=0.012, CI = [.112, .575]; not shown), all measures thought to rely on a place code. The correlation between slow FM and slow AM was not significant ($r = 0.199$, p=0.145, CI = [−0.07, .441]; *Figure 3*, panel D) nor was the correlation between slow FM and fast AM ($r = 0.021$, p=0.438, CI = [−0.246, .285]; not shown). The lack of a significant correlation between slow FM and AM could reflect the use of an additional time code in slow FM. However, because the correlation between slow FM and slow AM was not significantly different from the correlation between fast FM and fast AM ($Z = −0.906$, p=0.365, two-tailed), these differences in magnitudes should be interpreted with caution. Overall, the patterns of FM and AM correlations do not strongly support one hypothesis over another.

## Relationship between frequency selectivity and FM detection thresholds

The unitary neural coding theory of FM and AM predicts that steeper masking functions (implying sharper cochlear tuning) should be related to better FM detection thresholds (*Zwicker, 1956*). Furthermore, this relationship should hold even after controlling for central aspects of processing known to relate to FM detection, such as aging and sensitivity to AM at the same rate (*Whiteford and Oxenham, 2015*; *Whiteford et al., 2017*), as well as overall hearing loss for the tested ear, which could affect time coding independently of place coding (e.g., *Ewert et al., 2020*). The current consensus is that place theory applies to fast but not slow FM detection (*Moore and Sek, 1995*; *Moore and Sek, 1996*; *Lacher-Fougère and Demany, 1998*; *Strelcyk and Dau, 2009*). Our results contradict this consensus by showing that both slow and fast FM detection thresholds were strongly and similarly related to the masking function slopes (*Figure 4*). Notably, a few participants had masking function slopes of zero, or even in the opposite direction (i.e., negative low slopes and/or positive high slopes), presumably due to measurement noise. Imputing these 'opposite' slope values with 0 or removing these participants did not affect the statistical outcomes of the analysis. Age and sensitivity to AM could confound effects of cochlear filtering because they are both correlated with FM detection in listeners with normal hearing (*Whiteford and Oxenham, 2015*; *Paraouty et al., 2016*; *Whiteford et al., 2017*). Audibility is not thought to affect FM for levels that are 25 dB or more above absolute threshold (*Zurek and Formby, 1981*), but average absolute thresholds for the carrier frequency in the tested ear were included in the partial correlation analysis as a precaution, since a few listeners with the most hearing loss had stimuli presented at or near 20 dB sensation level (SL), and because hearing loss has been postulated to affect time coding, independent of place coding (*Ewert et al., 2020*). Partial correlations between FM detection and masking function slopes were conducted, controlling for age, absolute thresholds at 1 kHz, and AM detection thresholds at the corresponding rate, in an attempt to isolate the role of place coding in FM detection. The correlations between the residuals (*Figure 5*) demonstrate a significant relationship between the low slopes of the masking function and FM detection thresholds at both rates (slow FM: $r_p = −0.364$, p=0.016, CI = [−0.58, −0.101]; fast FM: $r_p = −0.377$, p=0.015, CI = [−0.589, −0.116]) but no relation between the high slope and FM (slow FM: $r_p = −0.064$, p=0.555, CI = [−0.331, .213]; fast FM: $r_p = −0.084$, p=0.555, CI = [−0.349, .194]). Because the low slope of the masking function (reflecting the upper slopes of the cochlear filters) is generally steeper than the high slope (e.g., *Figure 2—figure supplement 1*), it provides more information about frequency change than the high slope (*Figure 1*, leftmost column), and is therefore predicted to dominate FM performance (*Zwicker, 1956*). The correlations between the low slope and FM thresholds were significantly stronger than the correlations between the high slope and FM thresholds for both slow and fast FM before correcting for multiple comparisons (slow FM: $Z = −1.83$, p=0.034; fast FM: $Z = −1.81$, p=0.035) but not after (slow FM: p=0.065; fast FM: p=0.065), providing modest but mixed evidence that the low side of the excitation patterns matters most for FM-to-AM conversion. Sensitivity to AM detection was not related to either the low slopes (slow AM: $r = 0.058$, p>0.499, CI = [−0.211, .318]; fast AM:

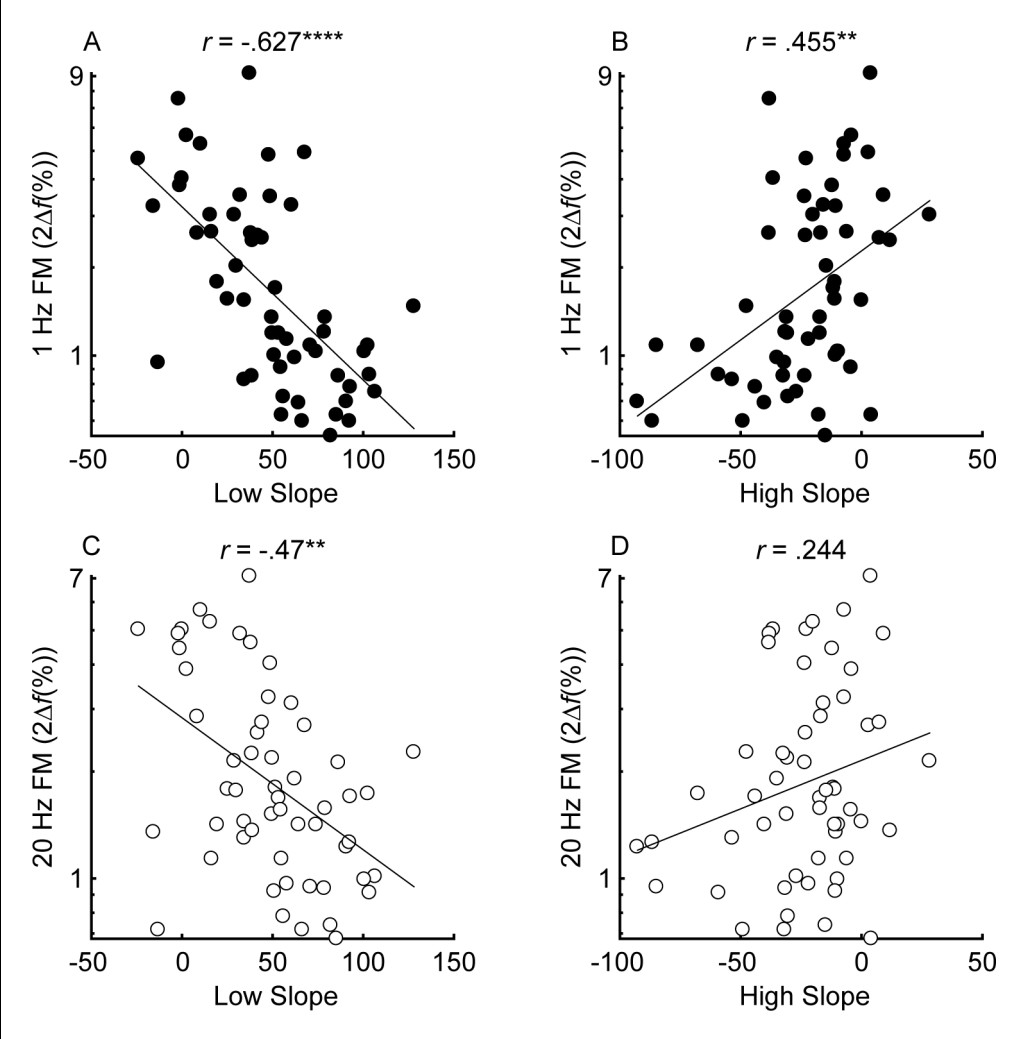

**Figure 4.** Correlations between the low slope (**A** and **C**) and high slope (**B** and **D**) and slow ($f_m$ = 1 Hz; black) and fast ($f_m$ = 20 Hz; white) FM detection (n = 55). Correlations marked with an * are significant after Holm's correction (****p<0.0001, ***p<0.001, **p<0.01, and *p<0.05).
The online version of this article includes the following source data and figure supplement(s) for figure 4:

**Source data 1.** *Figure 4* data.
**Figure supplement 1.** Frequency selectivity and FM detection correlations when both ears are included from subjects with asymmetric hearing loss (n = 66 ears).
**Figure supplement 1—source data 1.** Rows are individual subjects.
**Figure supplement 1—source data 2.** Rows are individual subjects.
**Figure supplement 2.** Relationship between place coding fidelity and FM sensitivity with outlier included.

$r$ = 0.277, p=0.076, CI = [.013, .505]) or the high slopes (slow AM: $r$ = 0.007, p>0.499, CI = [−0.259, .272]; fast AM: $r$ = −0.281, p=0.076, CI = [−0.508, −0.017]) of the masking functions. Importantly, the correlations between FM detection thresholds and the low slopes were significantly stronger than the correlations between AM and the low slopes (slow: $Z$ = −4.42, p<0.0001; fast: $Z$ = −4.89, p<0.0001), demonstrating that the relation between masking function slopes and modulation detection is specific to FM, as predicted by the place-coding theory (*Figure 1*). The results therefore provide strong support for the hypothesis that place coding is utilized for FM detection at both slow and fast rates.

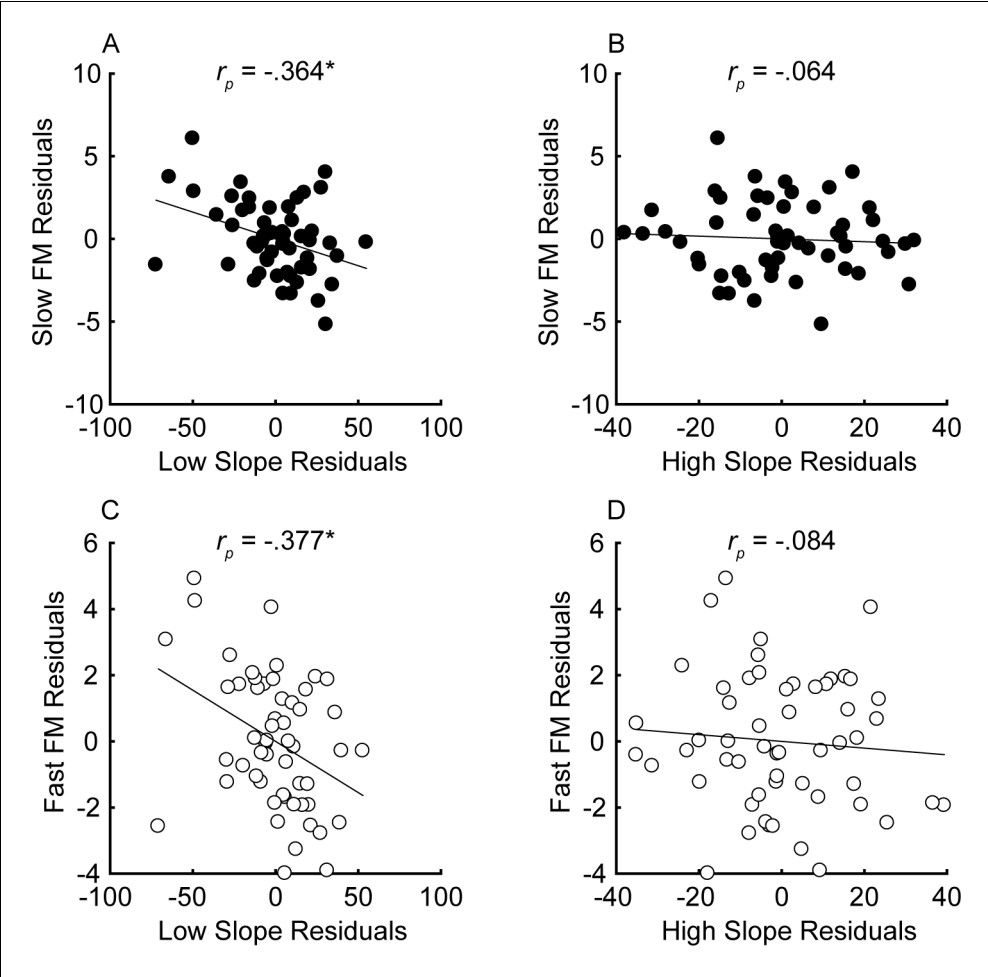

**Figure 5.** Partial correlations between the steepness of the masking function slopes (x-axis) and FM detection (y-axis) for slow (**A and B**) and fast FM (**C and D**) after variance due to audibility, sensitivity to AM, and age has been partialled out for n = 55 participants. Units of the x and y axes are arbitrary because they correspond to the residual variance for slow ($f_m$ = 1 Hz; black) and fast FM detection ($f_m$ = 20 Hz; white). Correlations marked with an * are significant after Holm's correction (****p<0.0001, ***p<0.001, **p<0.01, and *p<0.05).

The online version of this article includes the following source data and figure supplement(s) for figure 5:

**Source data 1.** *Figure 5* data.
**Figure supplement 1.** Partial correlations between place coding fidelity and FM sensitivity when both ears are included for asymmetric subjects (n = 66 ears) for slow (**A and B**) and fast FM (**C and D**).
**Figure supplement 1—source data 1.** Rows are individual subjects.
**Figure supplement 1—source data 2.** Rows are individual subjects.
**Figure supplement 2.** Partial correlations between place coding fidelity and FM sensitivity with outlier included for slow (**A and B**) and fast FM (**C and D**).

## Quantifying contributions from absolute thresholds, age, and sensitivity to AM

Multiple linear regression was conducted to determine how much variance each factor contributes to FM sensitivity. Unlike correlations, which are bi-directional, multiple linear regression is a conservative, directional approach to examining the amount of variance accounted for by each variable. Because many of the variables are correlated, the order the variables are entered into the model will affect the percentage of variance explained by each variable. The Variance Inflation Factors ranged from 1.1 to 3.69 for the slow FM model and 1.06–3.62 for the fast FM model, well below the common cutoff of 10, indicating that the independent variables are not too highly collinear with one

another (*Marquardt, 1970*). We took the most conservative approach by entering the low and high slopes last, after all the other variables. Factors known or believed to contribute to FM sensitivity (1 kHz absolute thresholds in the measured ear, age, sensitivity to AM at the corresponding rate, low slope, and high slope, entered in this order) were entered into the model, fitted using the Ordinary Least Squares method. The full models, with all variables entered, explained 59.5% (p<0.0001) and 52.1% (p<0.0001) of the variance in slow and fast FM, respectively. When sequentially entering each variable, absolute thresholds accounted for 45.2% (p=0.022) and 21.7% (p=0.11, n.s.) of the variance for slow and fast FM, respectively (note that all of the *p* values here correspond to the significance of the variable in the full model). Because age is known to impair FM detection (*He et al., 2007*; *Strelcyk and Dau, 2009*; *Grose and Mamo, 2012*; *Paraouty et al., 2016*; *Wallaert et al., 2016*; *Paraouty and Lorenzi, 2017*; *Whiteford et al., 2017*), age was entered into the model second, accounting for an additional 4.03% (slow FM: p=0.074, n.s.) and 3.78% (fast FM: p=0.16, n.s.) of the variance, while AM thresholds, entered third accounted for 4.03% (slow FM: p=0.039) and 18.7% (fast FM: p<0.0001). The low slope (slow FM: 6.2%, p=0.009; fast FM: 7.91%, p=0.008) but not the high slope (slow FM: .03%, p=0.864; fast FM: .01%, p=0.938) significantly contributed to the variance in sensitivity to FM at both rates, consistent with the partial correlation analysis (*Figure 5*). Note that entering the slopes first, instead of last, into the regression means that the variance explained is the same as the squared correlations plotted in panels A and C of *Figure 5* (e.g., slow FM and low slope: 39.3%; fast FM and low slope: 22.1% variance explained), but the total variance explained in the full models is unaffected. Entering just the low slope and AM at the corresponding rate into the MLRs, which would be consistent with Zwicker's (1956) place model, accounts for 44.9% and 43.7% of the variance for slow and fast FM, respectively, with significant contributions from both the low slope (slow FM: p<0.0001; fast FM: p<0.0001) and AM (slow FM: p=0.026; fast FM: p<0.0001). Overall, the results are consistent with a role of a place code in FM detection at both slow and fast rates.

## Experiment 2: Simulating FM through AM incoherence

The results from experiment 1 suggest that the fidelity of place coding relates to FM sensitivity, but they do not preclude the possibility that a time code is used that is dependent on frequency selectivity (e.g., *Shamma and Klein, 2000*). The strongest arguments in support of a time-interval code for slow-rate FM are (1) that FM sensitivity is higher (better) at slow rates than at fast rates (in contrast to AM sensitivity), possibly reflecting 'sluggishness' in the ability to evaluate phase-locked timing information, and (2) that sensitivity to slow-rate FM degrades at high carrier frequencies (*Sek and Moore, 1995*), where phase-locked timing information in the auditory nerve also degrades (*Johnson, 1980*; *Palmer and Russell, 1986*; *Verschooten et al., 2019*), whereas sensitivity to AM does not (*Kohlrausch et al., 2000*). Experiment 2 was designed to test whether these two properties of FM in normal-hearing listeners could also be explained via a place-coding mechanism. If FM is coded via an FM-to-AM place-based mechanism, then sensitivity to out-of-phase fluctuations in amplitude at nearby cochlear locations (as also produced by FM; see *Figure 6*, panels A and B) should be greater at slow than at fast fluctuation rates. We applied AM to two separate carriers, with the modulation either in phase (coherent) or out of phase (incoherent) between the two carriers, at fast and slow modulation rates, and with the carriers presented at frequencies that ranged from within to outside the putative range of human auditory-nerve phase locking (*Verschooten et al., 2018*). The wide spectral separation of the two carriers (2/3 and 4/3 octaves), their low sound level (45 dB SPL), as well as the narrowband noise added in the spectral gap between them, ruled out any peripheral interactions (including combination tones) between the carriers. We tested whether young normal-hearing listeners' ability to discriminate incoherent from coherent AM, as well as their ability to detect incoherent AM, varied in the same way that FM detection thresholds varied with modulation rate and carrier frequency, as would be predicted by the place-coding theory of FM perception.

### AM incoherence discrimination

Participants heard a sequence of three AM dyads. The task was to pick the dyad with components that had temporal envelopes 180° out of phase (*Figure 6D–E*). Carrier frequencies were presented either 2/3 (narrow frequency separation) or 4/3 (wide frequency separation) octaves apart, thereby simulating the effects of FM on the cochlear place representation, but without the presence of any

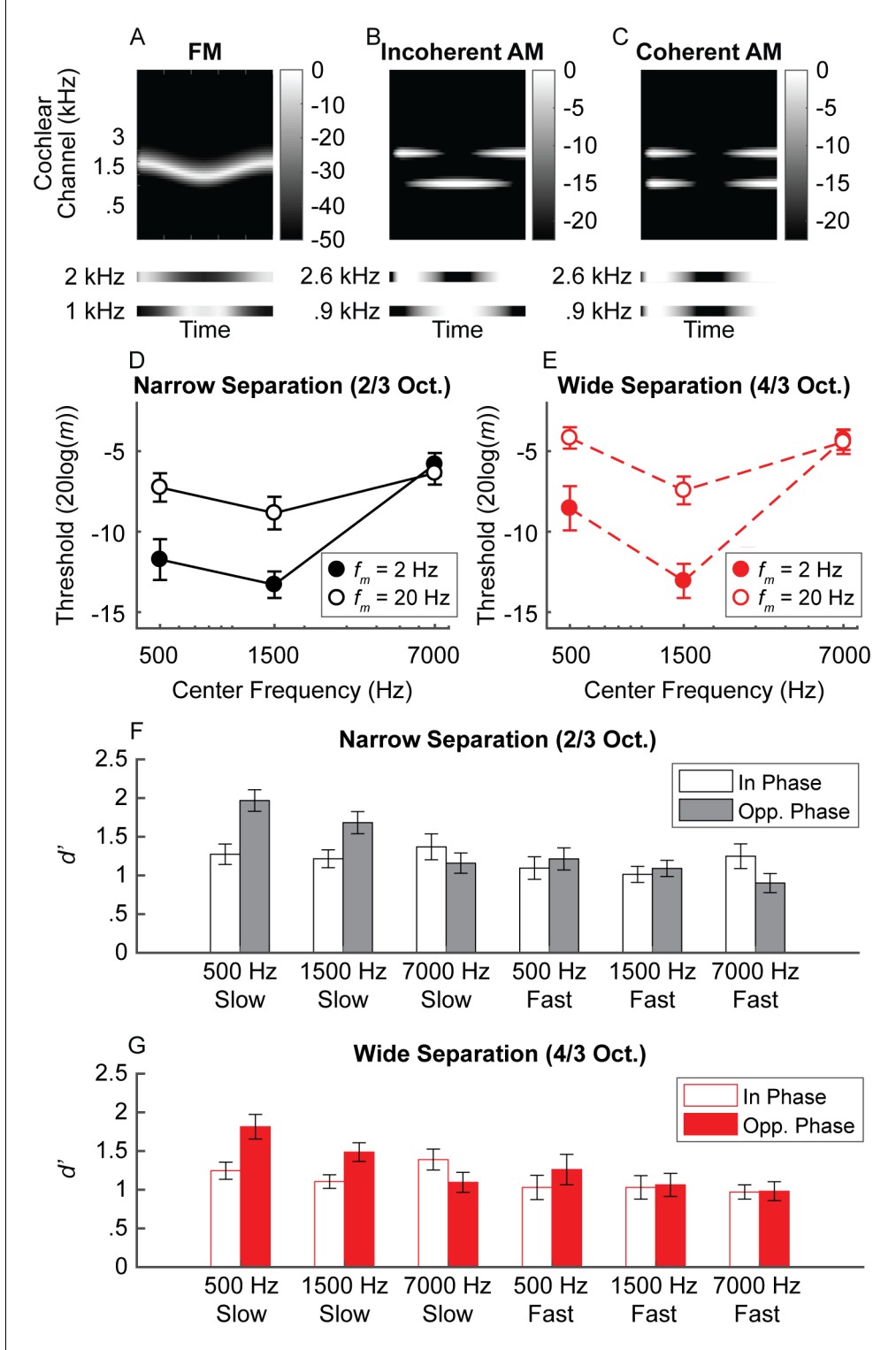

**Figure 6.** Experiment 2 schematic cochleagram and results. (**A**) Schematic cochleagram of an FM tone. The magnitude responses for two off-frequency filters (bottom) are 180-degrees out of phase for any given snapshot in time. (**B and C**) The schematic cochleagrams of two-component AM dyads with envelopes that are in opposite phase (incoherent) versus in phase (coherent). Incoherent envelopes lead to rate-place fluctuations similar to that observed in FM. (**D and E**) Average sensitivity for discriminating incoherent AM at slow (filled circles) and fast (open circles) rates in the narrow (black) and wide (red) frequency separation conditions. Sensitivity for simulated FM is best at lower center frequencies and slow rates, with slightly higher thresholds at very low center

*Figure 6 continued on next page*

*Figure 6 continued*

frequencies, similar to traditional FM sensitivity (*Sek and Moore, 1995*). (**F and G**) Sensitivity for detecting in-phase (open bars) and opposite-phase (filled bars) AM for two-component dyads. Sensitivity for opposite-phase AM (i.e., simulated FM) is only boosted for low center frequencies at slow rates, meaning a unified neural place code can account for limits in human FM sensitivity. N = 20 in all measures. Error bars represent ±1 standard error of the mean.

The online version of this article includes the following source data for figure 6:

**Source data 1.** *Figure 6* data.

informative fine timing cues produced by peripheral interactions between the two carriers. As predicted, sensitivity to incoherent AM mirrors trends seen in traditional FM sensitivity (center frequency × rate interaction: $F_{1.51,28.7}$ = 16.6, p<0.0001; *Figure 6*; *Supplementary file 1A*), with better performance at the slow than the fast rate for low (500 Hz: p=0.012; 1500 Hz: p=0.0002) but not high center frequencies (7000 Hz: p=0.648). The effect of rate on center frequency was not influenced by frequency separation ($F_{2,38}$ = .841, p=0.439), although there was an interaction between frequency separation and center frequency ($F_{2,38}$ = 11.6, p=0.0001). Performance was generally elevated in the wide condition, but more so at the 500 Hz and 7000 Hz center frequencies than at 1500 Hz ($p \leq$ .012 for all three narrow versus wide comparisons at each center frequency). The pattern of sensitivity to AM incoherence suggests that effects of carrier frequency and rate on modulation sensitivity are not unique to the phase-locked neural response to temporal fine structure.

## Complex AM detection

Detecting the presence of incoherent AM (simulated FM) is closer to the demands of FM detection than a discrimination task. A unitary place code for FM predicts that sensitivity to simulated FM via incoherent AM should be best in low-frequency regions (<4–5 kHz) and at slow modulation rates (<10 Hz). To test this hypothesis, we assessed our normal-hearing listeners' sensitivity for AM dyads that were either in phase (traditional AM) or in opposite phase (incoherent AM, simulating FM), with the modulation depth of each component set to 4 dB below each individual's AM detection threshold. Sensitivity was assessed for the same center frequencies, rates, and frequency separations as in the discrimination experiment. As predicted, the results revealed a significant three-way interaction between phase, center frequency, and rate (*Figure 6F–G*; *Supplementary file 1B*; phase × center frequency × rate: $F_{(2,38)}$ = 7.58, p=0.002). Sensitivity was greater for the opposite-phase conditions than for the in-phase conditions, but only when the center frequencies were low and the rate was slow (slow 500 Hz: p=0.0003; fast 500 Hz: p=0.132; slow 1500 Hz: p=0.01; fast 1500 Hz: p=0.501). At the highest center frequency, the slow-rate benefit for the opposite-phase condition was eliminated (slow 7000 Hz: p=0.132; fast 7000 Hz: p=0.132). This finding was not dependent on the amount of separation between the two carrier frequencies, as there was no significant main effect of frequency separation and no significant interactions. In summary, our detection tasks involving incoherent AM revealed the same pattern of results that are found in FM detection: performance was best at low carrier frequencies and slow modulation rates and was degraded at high modulation rates and/or high carrier frequencies.

# Discussion

## A unitary code for FM

Our finding that cochlear place coding is equally important for both slow- and fast-rate FM detection was unexpected. Humans' acute sensitivity to slow changes in frequency at carriers important for speech and music has been thought to result from precise neural synchronization to the temporal fine structure of the waveform or the combination of place and temporal fine structure cues (*Demany and Semal, 1989*; *Moore and Sek, 1995*; *Moore and Sek, 1996*; *Sek and Moore, 1995*; *Lacher-Fougère and Demany, 1998*; *Buss et al., 2004*; *Strelcyk and Dau, 2009*; *Johannesen et al., 2016*; *Paraouty et al., 2018*). Multiple linear regression analyses showed that the combined effect of audibility, age, sensitivity to AM, and masking function slopes accounted for about 60% and 52% of the total variance in slow and fast FM detection thresholds, respectively. This

is a high proportion of the variance, particularly considering the approximate and indirect nature of the behavioral measure used to estimate cochlear tuning.

The clear role for place coding in slow FM detection is contrary to the widely accepted view that time coding is used to detect FM at the slow rates found in speech and music. Instead, our results provide evidence for a unitary code for two crucial features of natural sounds, AM and FM, that extends across the entire range of naturally encountered fluctuation rates. Experiment 2 directly addresses the arguments that have been used in the past to support the use of a time-interval code or a dual place-time code for slow-rate FM and demonstrates that enhanced slow-rate FM sensitivity can be accounted for by limitations in humans' sensitivity to across-frequency variations in AM coherence, without recourse to the properties of auditory-nerve phase locking. This finding likely extends to simple frequency discrimination, which is believed to rely on the same mechanism as slow FM (*Sek and Moore, 1995*). A unitary code for FM and AM at all rates also helps explain the generally high-multicollinearity between FM and AM sensitivity observed here (*Figure 3*) and in several previous studies with normal-hearing listeners (*Whiteford and Oxenham, 2015*; *Otsuka et al., 2016*; *Paraouty and Lorenzi, 2017*; *Whiteford et al., 2017*), although the effect size of the correlation between slow-rate FM and AM was smaller than observed in previous studies. It may also help explain why attempts to improve speech and music perception by reintroducing fine timing cues via electrical pulses in cochlear implants have not been successful (*Zeng et al., 2005*; *Schatzer et al., 2010*).

## Implications for the perception and neural coding of complex tones

This study used pure tones, which are not frequently encountered in the natural environment. However, combinations of pure tones form harmonic complex tones, such as musical instrument sounds, voiced speech, and many animal vocalizations. It is known that humans perceive the pitch of harmonic complex tones in ways that are fundamentally different from other commonly studied species, such as the chinchilla (*Shofner and Chaney, 2013*), ferret (*Walker et al., 2019*), or songbird (*Bregman et al., 2016*). Recent work (*Shofner and Chaney, 2013*; *Walker et al., 2019*) has suggested that part of this difference can be explained by the substantially sharper cochlear tuning found in humans than in smaller mammals (*Shera et al., 2002*; *Shera et al., 2010*; *Sumner et al., 2018*; *Verschooten et al., 2018*). Specifically, sharper human cochlear tuning is believed to explain why humans (*Houtsma and Smurzynski, 1990*; *Bernstein and Oxenham, 2003*) and some other primates (*Song et al., 2016*) rely primarily on low-numbered spectrally resolved harmonics to extract pitch, whereas smaller mammals, such as ferrets and chinchillas, seem to rely primarily on the cues in the temporal envelope, which are provided by spectrally unresolved harmonics (*Shofner and Chaney, 2013*; *Walker et al., 2019*).

Results from the present study suggest that resolved harmonics, which are most important for human pitch perception, may be represented by their place of stimulation rather than by the temporal fine structure information encoded via the stimulus-driven spike timing (phase locking). This conclusion is consistent with other studies showing that pitch perception is possible even with spectrally resolved harmonics that are usually assumed to be too high in frequency to elicit phase locking (*Oxenham et al., 2011*; *Lau et al., 2017*; *Carcagno et al., 2019*) and with studies showing that steep filter slopes are required to represent harmonics from filtered noise in noise-vocoder simulations (*Mehta and Oxenham, 2017*). It is also supported by recent data from the inferior colliculus of the rabbit, showing that place coding of low-numbered harmonics from high F0s in the midbrain is robust over a relatively wide range of sound levels (*Su and Delgutte, 2020*), a finding that should generalize to low F0s in humans, given our superior frequency selectivity.

## Alternative interpretations

One alternative interpretation of our results from experiment 1 is that hearing loss leads to a degradation in both spectral resolution and neural phase locking to temporal fine structure, and that it is the degradation in the phase locking, not cochlear filtering, that drives the relationship between spectral resolution and FM coding observed here. There are several reasons why this interpretation is unlikely. First, physiological studies with non-human animals have generally found little or no effect of SNHL on phase locking in the auditory nerve (*Harrison and Evans, 1979*; *Miller et al., 1997*; *Henry and Heinz, 2012*; *Henry et al., 2019*), with the exception of one study (*Woolf et al., 1981*).

Support from human studies is based on indirect evidence from behavioral results showing poorer performance in hearing-impaired listeners in tasks thought to use time coding (*Lorenzi et al., 2006*; *Moore et al., 2006*; *Moore et al., 2012*; *Hopkins and Moore, 2007*; *Hopkins and Moore, 2011*; *Moore, 2014*; *Füllgrabe and Moore, 2017*). However, all of these, with the exception of binaural tasks, could be affected by poorer cochlear tuning (*Oxenham et al., 2009*). Binaural tasks, involving the discrimination of interaural time differences (ITDs) in the temporal fine structure of stimuli, are likely to rely on phase-locked coding. These studies have not always found a clear relationship between ITD sensitivity and hearing loss, once effects of age and audibility are accounted for (e.g., *Smoski and Trahiotis, 1986*; *Hopkins and Moore, 2011*), although a recent meta-analysis suggests a small effect of hearing loss on ITDs once controlling for age (*Füllgrabe and Moore, 2018*).

A second reason why it is unlikely for the role of place coding in FM to be a byproduct of degraded time coding with SNHL is that the relationship between place coding fidelity and FM sensitivity remained significant even after the effects of age, AM sensitivity, and hearing loss at 1 kHz were accounted for via partial correlation. Controlling for hearing loss should ensure that possible confounding changes in temporal fine structure coding with poorer frequency tuning are factored out, whereas controlling for aging and sensitivity to AM accounts for central aspects of processing that may affect FM thresholds, including task demands (i.e., the ability to perform a two-interval modulation detection task) and processing efficiency (*Whiteford and Oxenham, 2015*; *Whiteford et al., 2017*). The results therefore suggest that degradations of time coding with hearing loss or central aspects of aging do not fully account for the observed effects.

Another alternative interpretation is that a dual code, based on combined place and timing cues, accounts for slow FM sensitivity, rather than a unitary code. A dual code could potentially explain the high collinearity often observed between measures of FM and AM sensitivity (*Whiteford and Oxenham, 2015*; *Otsuka et al., 2016*; *Paraouty and Lorenzi, 2017*; *Whiteford et al., 2017*), as well as the observation from experiment one that slow-rate FM sensitivity may not be as strongly correlated to AM sensitivity as fast-rate FM (although these differences in correlation strength were not statistically significant). In addition, it has been found that AM can interfere with the detection of FM, particularly at fast rates and high carrier frequencies, perhaps pointing to the possibility of a dual code (*Moore and Sek, 1996*; *Ernst and Moore, 2010*; *King et al., 2019*). However, just as experiment 2 showed that timing cues are not necessary to explain enhanced slow-rate, low-carrier FM sensitivity, it may be that coherent AM across two carriers also interferes with the detection of incoherent AM in the same way that it interferes with FM detection. This prediction remains to be tested.

## Explaining superior FM perception at low rates and low carrier frequencies within a unitary framework

A pure cochlear place-based model for FM proposes that FM is transduced to AM via cochlear filtering (*Zwicker, 1956*). As the frequency sweeps across the tonotopic axis, this is reflected via periodic amplitude fluctuations in the responses of cochlear filters. Experiment 2 demonstrated that a place-only model can account for the different rate- and frequency-dependent trends in FM and AM sensitivity observed in many previous studies (*Viemeister, 1979*; *Sheft and Yost, 1990*; *Moore and Sek, 1995*; *Moore and Sek, 1996*; *Sek and Moore, 1995*; *Lacher-Fougère and Demany, 1998*; *Moore and Skrodzka, 2002*; *Whiteford and Oxenham, 2015*; *Whiteford et al., 2017*; *Whiteford and Oxenham, 2017*), based on limitations in sensitivity to AM incoherence at high carrier frequencies and/or high modulation rates. Previous studies examining sensitivity to AM incoherence had either only tested fast rates (*Green et al., 1990*) or lower center frequencies (*Moore and Sęk, 2019*). *Moore and Sęk, 2019* noted that the very large AM depths needed to discriminate AM incoherence, also observed here, are larger than one might expect if such a task were reflective of the same mechanism used in FM coding. However, the carriers used in AM-incoherence experiments must be spaced far enough apart to avoid peripheral interactions (in our case 2/3 or 4/3 octaves), meaning that the separation is much greater than for the two sides of excitation produced by a single carrier in an FM experiment. This in itself could explain why overall sensitivity is poorer in the AM simulations than in true FM detection and discrimination experiments.

Our combined findings suggest the auditory system's ability to compare changes in the output between nearby cochlear filters is more efficient at slow than at fast rates, but only at low center frequencies. This interpretation is partly supported by a computational modeling study showing that

frequency and intensity can be represented by a single code, if inter-neuronal noise correlations (*Cohen and Kohn, 2011*) are taken into account (*Micheyl et al., 2013*). Such correlations would require relatively long time windows to play a functional role, and so would only provide a benefit at low modulation rates, where the duration of the necessary time window does not exceed one period of the modulation. It is not currently known why such effects are dependent on the carrier frequency, but it may be that auditory cortical representations of the highest frequencies (>6 kHz) may be less extensive, due to their relative unimportance for everyday auditory stimuli, such as speech and music, which in turn could produce poorer sensitivity in fine discrimination tasks, analogous to the effects of visual crowding observed in the visual periphery (e.g., *Whitney and Levi, 2011*).

## Materials and methods

### Experiment 1

#### Participants

All tasks in experiment one were completed by 56 participants (19 male, 37 female; average age of 66.5 years, range: 19.4–78.5 years) with no reported history of cognitive impairment. All participants underwent audiometric screening, involving air- and bone-conduction audiometric threshold measurements at octave frequencies between 250 and 8000 Hz. Nine participants had clinically normal hearing at the test frequency of 1 kHz (audiometric thresholds $\leq$ 20 dB hearing level, HL) in both ears. The other 47 participants had varying degrees of SNHL, with audiometric thresholds at 1 kHz poorer than 20 dB HL in at least one ear and air-bone gaps < 10 dB to preclude a conductive hearing loss. Psychoacoustic measurements of absolute threshold for a 500 ms 1 kHz tone resulted in thresholds ranging from −0.7 to 68.5 dB SPL. Ears with thresholds of 70 dB SPL or more at 1 kHz were not included in the study. Participants with symmetric hearing (n = 37; difference in absolute thresholds at 1 kHz $\leq$10 dB) completed all experimental tasks using the ear with the higher threshold at 1 kHz. Six participants had SNHL at 1 kHz in both ears, but loss in the poorer ear exceeded the study criterion; for these subjects, tasks were completed in the better ear only. One additional participant was only assessed in their better ear because loss in the poorer ear was near the study criterion (68.6 dB SPL at 1 kHz), and the subject indicated the sound level was uncomfortable. An additional three participants had one normal ear and one ear with SNHL at 1 kHz, and only measurements from the impaired ear were used in analyses. The final nine participants had asymmetric SNHL in both ears, defined as a difference in absolute thresholds > 10 dB at 1 kHz. For eight of these subjects, the experimental tasks were completed for both ears separately. One participant with asymmetric hearing only completed tasks in their poorer ear due to time constraints (*Table 1*). However, only performance in the poorer ear was used in the analyses for all nine of these listeners (see *Figures 4–5* – *Figure 4—figure supplement 1* and *Figure 5—figure supplement 1* for both ears included from all asymmetric listeners). Participants provided informed consent and were compensated with hourly payment or course credit for their time. The Institutional Review Board of the University of Minnesota approved all experimental protocols.

#### Stimuli

Stimuli were generated within Matlab (MathWorks) at a sampling rate of 48 kHz using a 24-bit Lynx Studio L22 sound card and were presented via Sennheiser HD650 headphones to participants individually seated in a sound-attenuating booth. The test stimuli were presented monaurally with

**Table 1.** Summary of participants.

| Measured ear | # of Participants | Notes |
|---|---|---|
| Worse ear | 38 | Subjects with symmetric 1 kHz thresholds (asymmetry <= 10 dB; n = 37) or who could only be assessed in their worse ear due to time constraints (n = 1). |
| Better ear | 7 | Subjects with 1 kHz thresholds in the worse ear that exceeded the study criterion (n = 6) or indicated the SL in their worse ear was uncomfortable (n = 1). |
| Both ears (worse ear used in analyses) | 11 | 1 kHz asymmetry > 10 dB; n = 3 had normal hearing in their better ear, and n = 8 had SNHL in both ears. |

threshold equalizing noise (TEN; *Moore et al., 2000*) presented in the contralateral ear to prevent audible cross-talk between the two ears. The TEN was presented continuously in each trial, with the bandwidth spanning one octave, geometrically centered on the test frequency. Except for tasks that involved detection of a short (20 ms) tone pip, the TEN level (defined as the level with the auditory filter's equivalent rectangular bandwidth at 1 kHz) was always 25 dB below the target level, beginning 300 ms before the onset of the first interval and ending 200 ms after the offset of the second interval. Because less noise is needed to mask very short targets, the TEN was presented 35 dB below the target level for tasks that involved detection of a short, 20 ms tone pip (with and without the presence of a forward masker). This noise began 200 ms before the onset of the first interval and ended 100 ms after the offset of the second interval.

To obtain a more precise estimate of sensitivity for the test frequency, pure-tone absolute thresholds were measured for each ear at 1 kHz. The target interval contained a 1 kHz, 500 ms tone with 10 ms raised-cosine onset and offset ramps. The reference interval was 500 ms of silence, and the target and reference intervals were separated by a 400 ms interstimulus interval (ISI). Tasks involving modulation detection were assessed for the same frequency ($f_c$ = 1 kHz) at slow ($f_m$ = 1 Hz) and fast ($f_m$ = 20 Hz) modulation rates. The target was a pure tone with FM or AM while the reference was an unmodulated pure tone at 1 kHz. Both the target and the reference tones were 2 s in duration with 50 ms raised-cosine onset and offset ramps. In the FM tasks, the starting phase of the modulator frequency was set so that the target always began with either an increase or decrease in frequency excursion from the carrier frequency, with 50% a priori probability. A similar manipulation was used for the AM tasks, so that the target always began at either the beginning or middle of a sinusoidal modulator cycle and so was either increasing or decreasing in amplitude at the onset. Stimuli for the modulation tasks were presented at 65 dB SPL or 20 dB sensation level (SL), whichever was greater, based on the individual participant's absolute thresholds at 1 kHz.

Detection thresholds for a 20 ms tone pip were measured with and without the presence of a 1 kHz, 500 ms pure-tone forward masker. Tone-pip frequencies were 800, 860, 920, 980, 1020, 1080, 1140, and 1200 Hz, and both the tone pip and the masker had 10 ms raised cosine onset and offset ramps. The tone pip was presented to one ear, directly following the offset of the masker (0 ms gap), and the masker was presented to both ears to avoid potential confusion effects between the offset of the masker and the onset of the tone pip (*Neff, 1986*). The masker was fixed in level at either 65 dB SPL or 20 dB SL, whichever was greater, based on absolute thresholds for the 500 ms 1 kHz tone in the target ear. For the unmasked conditions, the tone pip was preceded by 500 ms of silence.

## General procedures

All experiments were created within the AFC software package (*Ewert, 2013*) in Matlab. Procedures were adapted from *Whiteford et al., 2017*. The experiment took place across 3–6 separate sessions, with each session lasting no longer than 2 hr. All tasks were carried out using a two-interval, two-alternative forced-choice procedure with a 3-down 1-up adaptive method that tracks the 79.4% correct point of the psychometric function (*Levitt, 1971*). The target was presented in either the first or second interval with 50% a priori probability, and the participant's task was to click the virtual button on the computer screen (labeled '1' or '2') corresponding to the interval that contained the target. Each corresponding response button illuminated red during the presentation of the stimulus (either reference or target). Visual feedback ('Correct' or 'Incorrect') was presented on the screen after each trial. All participants completed the tasks in the same order, and the tasks are described below in the order in which they were completed by the participants.

## Absolute thresholds at 1 kHz

Participants were instructed to select the button on the computer screen that was illuminated while they heard the 500 ms 1 kHz tone. Three runs were measured for each ear, and the order of the presentation ear (left versus right) was randomized across runs. Three participants were only assessed in their better ear, due to the extensive hearing loss in the poorer ear according to their 1 kHz audiometric thresholds (all ≥80 dB HL). The remaining participants completed monaural absolute thresholds for both ears.

On the first trial, the target was presented at 40 dB SPL. The step size for the adaptive procedure was 8 dB up to the first reversal, 4 dB for the next two reversals, and 2 dB for all six following reversal points. Absolute thresholds were determined by calculating the mean level at the final six reversal points. If the standard deviation (SD) across the three runs was ≥4 dB, then three additional runs were conducted for the corresponding ear, and the first three runs were regarded as practice.

### FM detection

Participants were instructed to select the interval that contained the tone that was 'modulated' or 'changing'. At the beginning of each run, the target had a peak-to-peak frequency excursion ($2\Delta f$) of 5.02%. The excursion varied by a factor of 2 for the first two reversal points, a factor of 1.4 for the third and fourth reversal points, and a factor of 1.19 for the final six reversal points. The FM difference limen (FMDL) was defined as the geometric mean of $2\Delta f$ at the final six reversal points.

Three runs were conducted for each modulation rate, and all three runs for slow-rate FM ($f_m = 1$ Hz) were completed before fast-rate FM ($f_m = 20$ Hz). Participants with asymmetric hearing loss at 1 kHz who had two qualifying ears completed six runs (three runs per ear) for each modulation rate, and the order of the presentation ear was randomized across runs. If the SD across the three runs for a given ear was ≥4 in units of $10\log(\Delta f(\%))$, the participant completed an additional three runs, and only the last three runs were used in analyses.

### Detection of 20 ms tones in quiet

Participants were instructed to select the button (labeled '1' or '2') on the computer screen that was illuminated while they heard a short, 20 ms target tone pip. The target was presented at 40 dB SPL or 20 dB SL, whichever was greater, for the first trial of each run. The initial step size for the target level was 8 dB for the first two reversals, 4 dB for the following two reversals, and 2 dB for the final six reversals. The absolute threshold was defined as the mean target level at the final six reversal points.

Participants completed one run for each of the eight tone-pip frequencies: 800, 860, 920, 980, 1020, 1080, 1140, and 1200 Hz. The order of the tone-pip frequency conditions was randomized between runs and between participants. Participants with asymmetric hearing loss and two qualifying ears had the order of the runs further blocked by presentation ear, so that eight runs for the same ear had to be completed before any conditions in the opposite ear were measured. Whether the right or left ear was assessed first was randomized between participants. One additional run was conducted for any run that resulted in a SD ≥4 dB for the tone-pip levels at the final six reversal points, and only the final run for each condition was used in analyses.

### AM detection

The instructions for AM detection were the same as the instructions for FM detection. The first trial of each run had a target with an AM depth of −8 dB, in units of $20\log(m)$, where $m$ is the modulation index (from 0 to 1). The target modulation depth changed by 6 dB for the first two reversals, 2 dB for the next two reversals, and 1 dB for the final six reversals. The AM difference limen (AMDL) was defined as the mean modulation depth at the last six reversal points.

In the same manner as for the FM tasks, all three runs for slow-rate AM ($f_m = 1$ Hz) were completed before the fast-rate AM ($f_m = 20$ Hz) runs. Participants with asymmetric hearing loss at 1 kHz and two qualifying ears completed six runs (three runs per ear) for each modulation rate, and the order of the presentation ear was randomized across runs. If the SD of the threshold estimates from the first three runs for a given condition were ≥4 dB, then three additional runs were conducted, and only the final three runs were analyzed.

### Forward masking patterns

The task was to determine which of two tones was followed by a short, 20 ms tone pip. Two runs were measured for each of the eight tone-pip frequencies (800, 860, 920, 980, 1020, 1080, 1140, and 1200 Hz), for a total of 16 runs, and the order of the tone-pip conditions was randomized across runs. Participants with asymmetric hearing loss at 1 kHz and two qualifying ears had the order of the runs further blocked by presentation ear, so that eight runs for the same ear had to be completed before any conditions in the opposite ear were presented. Within a trial, each masker was either

directly followed by a 20 ms tone pip, presented monaurally to the target ear, or 20 ms of silence. The starting level of the tone pip was 10 dB below the masker level in the corresponding ear. The level of the tone pip changed by 8 dB for the first two reversals, 4 dB for the third and fourth reversals, and 2 dB for the following six reversals. The masked threshold for each tone-pip frequency condition was calculated as the mean tone-pip level at the final six reversal points. For a given subject, if the SD of the masked threshold across the two runs was $\geq$4 dB, then the subject completed two additional runs for the corresponding tone-pip frequency. For these conditions, only the final two runs were used in analyses, and the first two runs were regarded as practice. The average across the final two runs for each tone-pip frequency was used in analyses.

## Sample size

Because the strength of the relationship between FM sensitivity and forward masking slopes was unknown in listeners varying in degree of SNHL, and the number of people with SNHL at 1 kHz was expected to be limited, we set a minimum sample size requirement for SNHL subjects based on the smallest effect we would like to be able to detect. To detect a moderate correlation between masking function slopes and FM sensitivity ($r$ = 0.4, $\alpha$ = 0.05, one-tailed test) with a power of .9, we needed a sample of n = 47. We also aimed to recruit an additional 10 participants with normal hearing thresholds at 1 kHz of similar age to the SNHL subjects. The normal-hearing sample was limited to 10 participants to ensure a relatively even distribution of absolute thresholds at 1 kHz between 0 and 70 dB SPL. One of these anticipated normal-hearing subjects had mild SNHL at 1 kHz in their worse ear, leading to a sample size of n = 57, with nine listeners with normal hearing at 1 kHz and 48 listeners with SNHL. One SNHL subject reported a history of neurological issues and was excluded from the study. Another SNHL subject had unusually poor FM sensitivity at both rates, with thresholds greater than 3 SD from the group mean. This outlier was excluded from all analyses, leading to a final sample size of n = 55. Including the outlier in all analyses generally did not affect the results (see *Supplementary file 1C* and *Figures 4–5 – Figure 4—figure supplement 2*, *Figure 5—figure supplement 2*).

## Experiment 2

### Participants

Twenty participants (three male, 17 female; mean age = 21.8 years; range: 19–28 years) completed the full experiment 2. An additional 18 participants began the study but failed one or more of the screening criteria: One participant failed the audiometric screening, two participants failed the absolute threshold screening at one or both frequencies, and 15 participants were unable to pass the AM discrimination screening. The large number of participants unable to pass the AM discrimination screening was likely due to the limited amount of training provided. Most participants were experienced with psychophysical tasks and were recruited from the laboratory's participant pool. To pass the audiometric screening, participants were required to have pure-tone thresholds $\leq$ 20 dB HL at octave frequencies between 250 and 8000 Hz. All participants gave informed consent and were provided monetary compensation or course credit for their time. All protocols were approved by the Institutional Review Board of the University of Minnesota.

### Stimuli

Stimuli were generated digitally in Matlab (MathWorks) at a sampling rate of 48 kHz with a 24-bit Lynx E22 soundcard and were presented diotically over Sennheiser HD 650 headphones in a double-walled sound-attenuating booth. Absolute thresholds in quiet were assessed for the two highest frequency components present during the experiment (8819 and 11112 Hz). The target interval contained a 500 ms pure tone with 10 ms raised cosine onset and offset ramps, while the reference interval contained 500 ms of silence. The target and reference intervals were separated by a 100 ms ISI.

For all other tasks, the stimuli were 1 s in duration with 50 ms raised cosine onset and offset ramps, presented at 45 dB SPL per component. AM discrimination was assessed for two carrier frequencies separated by either 2/3 (narrow-frequency separation) or 4/3 (wide-frequency separation) octaves and centered on one of three possible frequencies: 500, 1500, and 7000 Hz. The frequencies of each carrier for all conditions are presented in *Supplementary file 1D*. Both carriers were

amplitude-modulated at either a slow (2 Hz) or fast (20 Hz) rate. The starting phase of the modulator was randomized for each stimulus presentation and were either in phase (same starting phase) or out of phase (180-degree phase shift) for the two carriers. Randomizing the envelope starting phase in this manner ensured that participants could only use the relationship between the two modulators, rather than the starting phase of either of the modulators alone, to perform the task. The target stimulus was always incoherent AM (i.e., 180-degree phase difference), while the two reference stimuli were coherent AM (i.e., in phase). To prevent participants from possibly using envelope cues in off-frequency filters (i.e., by monitoring fluctuations in output at filters centered between the two carriers, which could be systematically different for the coherent versus incoherent conditions), narrow-band TEN was geometrically centered between the two carrier frequencies with a bandwidth of either 1/6 octave (for the narrow frequency-separation condition) or 1/3 octave (for the wide frequency-separation condition) and was presented at 39 dB SPL per ERB. The TEN was presented continuously and gated between trials, beginning 300 ms before the onset of the first stimulus within a trial and ending 200 ms after the offset of the last stimulus. Example trials were presented at large depths ($m = 1$ and $m = 0.75$) for the 1500 Hz center frequency in the narrow-frequency separation condition at both rates, as these were the conditions where envelope incoherence was most salient. The AM-phase discrimination screening tested all combinations of conditions for the wide-frequency separation, as this was predicted to be more challenging than the narrow-frequency separation, effectively eliminating participants who would require more training to perform the task.

Pure-tone AM detection thresholds were measured for each individual component in the presence of the corresponding TEN for all carrier/TEN combinations used in the complex AM detection task for both the 2 and 20 Hz rates.

Complex AM detection sensitivity for both coherent and incoherent AM was assessed using the same center frequencies, frequency separations, and modulation rates as used in the AM discrimination task. The target was a two-component complex tone with AM imposed on both components while the reference was a steady (unmodulated) two-component complex tone. For each block, the frequency components of the reference were identical to the carrier frequencies in the target. The modulation depth of the target components was individualized to be 4 dB below the participant's average pure-tone AM detection threshold (in $20\log(m)$) for each individual component at the same carrier and rate in the presence of TEN. This depth was small enough to make the task challenging but avoided performance that was at chance level. TEN was presented in the same manner as in the AM discrimination task.

## General procedures

Both the high-frequency absolute threshold and the AM discrimination screenings took place on the first session. Only participants who passed both screenings were invited to complete the rest of the experimental tasks, which took place across an additional 5–6 sessions, with each session lasting no longer than 2 hr. All tasks were either two- or three-interval alternative forced choice, with the target appearing in each interval with equal a priori probability. The task was to click the numbered virtual response button on the computer screen that corresponded to the interval that contained the target. Visual feedback ('Correct' or 'Incorrect') was presented on the screen after each trial. All participants completed the tasks in the same order, described below in the order they were presented.

## Absolute threshold screening

To ensure performance in the highest center-frequency condition was not limited by audibility, participants were required to have absolute thresholds $\leq 30$ dB SPL for the two highest frequency components present during the experiment. Participants were instructed that each of the three response buttons on the screen would illuminate red, one at a time, and their task was to select the button that was illuminated while they heard a tone. The target level varied adaptively using a 2-down, 1-up adaptive method that tracked the 70.7% correct point of the psychometric function. Two runs were measured for each frequency condition. The order of the frequency condition was randomized across runs with the constraint that both frequencies were tested once before repeating a frequency condition.

On the first trial, the target was presented at 40 dB SPL. The target changed by 8 dB for the first reversal, 4 dB for the next two reversals, and 2 dB for all following reversals. Absolute thresholds for

each run were determined by calculating the mean level at the final six reversal points. Participants with average thresholds across the two runs >30 dB for either frequency condition were excluded from participating in the rest of the study.

## AM-phase discrimination screening

An additional qualification criterion was that participants had to be able to perform the discrimination task by the end of the first 2 hr session. First, to aid in target identification, participants were presented with two blocks of eight example trials each (four trials per depth). A trial consisted of three tones, presented sequentially over time, and participants had to pick the one that was incoherently modulated. The target and references always had a large modulation depth of either $m = 1$ or $m = 0.75$ so that the target envelope incoherence was salient. The examples were blocked by modulation rate, and the depth condition was randomized within a block. The order of the blocks was randomized. Participants were allowed to repeat up to three blocks of examples per condition.

The examples were followed by the AM-phase discrimination screening. The instructions were the same as the example trials, but the AM depth of all stimuli varied adaptively with performance using a 2-down, 1-up method to track the 70.7% correct point of the psychometric function (*Levitt, 1971*). Two runs were measured for each condition, for a total of 12 runs. The order of the runs was randomized by center frequency and then modulation rate, so that both modulation rates were tested before continuing to the next center frequency condition. Randomization was constrained so that all conditions were tested once before any conditions were repeated.

The initial trial in each run had a modulation depth (in $20\log(m)$) of $-8$ dB. The depth changed by 6 dB for the first two reversals, 2 dB for the next two reversals, and 1 dB for all following reversals. Threshold was defined as the average log-transformed modulation depth at the final six reversal points. To account for inability to perform the task, the tracking procedure terminated early if the maximum possible modulation depth (0, 100% modulation) was reached 15 times within a run. Any conditions with at least one failed threshold estimate were repeated until the participant successfully achieved two consecutive threshold estimates for the corresponding condition(s). If the end of the 2 hr session was reached and a participant was still unable to achieve a threshold for one or more conditions, they were excluded from participating in the full study.

## AM phase discrimination

Instructions and methods were the same as the AM discrimination screening, except that both the wide- and narrow-frequency separation conditions were tested. The order of the conditions was randomized by center frequency, then modulation rate, and then frequency separation, so that both frequency-separation conditions were tested for a given center frequency and rate before the next rate was tested for the same center frequency. There were four runs per condition, for a total of 48 runs. All conditions were tested once before any conditions were repeated.

Eight subjects were unable to converge on a threshold for at least one run. In all instances, this corresponded to the 7000 Hz center frequency, usually in the wide frequency-separation condition and the fast rate. One additional run was collected to replace each failed run so that all participants had four thresholds estimates per condition.

## AM detection

Two tones were presented sequentially in time, and participants were instructed to pick the one that was modulated. There were two runs per condition, for a total of 48 runs. The order of the runs was randomized by carrier frequency and then modulation rate, so that both rates were tested before the next carrier frequency was presented. All carrier frequency and rate combinations were tested before any conditions were repeated.

The modulation depth varied based on performance using a 3-down, 1-up adaptive method to estimate the 79.4% correct point on the psychometric function (*Levitt, 1971*). At the beginning of each run, the target had a modulation depth of $-8$. The depth varied by 6 dB for the first two reversals, 2 dB for the next two reversals, and 1 dB for all following reversals. AMDLs were calculated as the average modulation depth at the final six reversal points.

## Complex AM detection

Participants were presented with two complex tones, one at a time, and instructed to pick the tone that was modulated. To cue participants to listen for the correct modulation rate, each block began with five practice trials with larger modulation depths. The target in the first practice trial was presented 3 dB above threshold and then decreased in depth by 1 dB for each additional practice trial. Practice trials were immediately followed by 50 experimental trials, with the target component depths set to 4 dB below the individualized AM detection thresholds. At the start of this task, participants were informed that the modulation depths were individualized to be difficult but not impossible, so they should use the practice trials to help them identify what rate to listen for in the corresponding block. Blocks were randomized by center-frequency condition and then modulation rate, so that one block of each rate and frequency separation were completed before the participant was presented with the next center-frequency condition. After completing one block of each condition, the randomization procedure was repeated again, so that participants completed 100 trials per condition.

## Sample size

Because the variance of AM incoherence sensitivity is unknown, sample size was determined by reviewing recent publications comparing group-average FM detection thresholds at different carriers and rates (*Moore and Sek, 1996*; *Ernst and Moore, 2010*; *King et al., 2019*). Our sample size of n = 20 is double that used in *King et al., 2019*, more than a factor of 3 of that used in *Ernst and Moore, 2010*, and more than a factor of 6 greater than *Moore and Sek, 1996*. A larger sample size was used under the assumption that the variance would be greater for simulated FM compared to traditional FM. There were no outliers for either AM phase discrimination or complex AM detection, defined as individuals with average performance that was ±3 standard deviations from the group average, so all 20 participants were included in analyses.

## Statistical analyses

Either $d'$ or mean log-transformed thresholds [$10\log_{10}(2\Delta f (\%))$ and $20\log_{10}(m)$] were used in all analyses, where $2\Delta f (\%)$ is the peak-to-peak frequency excursion (for FM) as a percentage of the carrier frequency, and $m$ is the modulation index (for AM). All reported means ($\bar{x}$) and standard deviations ($s$) for thresholds correspond to the log-transformed data. Confidence intervals (CIs) are 95%. Analyses were conducted using Matlab 2016b, IBM SPSS Statistics 25, and R (*R Development Core Team, 2019*). All statistical tests are one-tailed unless otherwise stated.

## Experiment 1

Pearson correlations were used to assess continuous trends; the corresponding $p$ values were adjusted using Holm's method to correct for family-wise error rate (*Holm, 1979*) implemented with the 'stats' package in R (*R Development Core Team, 2019*). The $p$ values corresponding to the correlations were corrected for two comparisons for *Figures 1*, six comparisons for *Figure 3* (all FM and AM correlations), and eight comparisons for *Figures 4* and *5* (all FM correlations with masking function slopes). The masking function slopes and AM correlations were corrected for four comparisons. The cocor package was used to calculate significant differences between correlations using Steiger's modification of Dunn and Clark's Z (*Dunn and Clark, 1969*; *Steiger, 1980*; *Diedenhofen and Musch, 2015*).

Bootstrap analyses were conducted to estimate the highest possible correlation detectable for each modulation task and the forward masking task, in order to ensure that correlations with these measures were not limited by test-retest reliability. For each subject and for each modulation condition, performance was simulated by randomly sampling six runs (three test and three retest) from a normal distribution based on the individual means and standard deviations from the corresponding task. An analogous procedure was conducted for each individual's masked thresholds for every tone-pip condition, with four runs (two test and two retest) sampled from each individualized normal distribution. The average simulated runs were used to estimate the low and high frequency slopes of the masking function by calculating a linear regression between the four lowest and four highest tone-pip frequency conditions for the average test and the average retest runs (four regressions per iteration). Simulated test-retest correlations were calculated using the simulated slopes for n = 55

subjects (for forward masking) or the simulated average test and retest thresholds for each subject (for the modulated tasks). This process was repeated for 100,000 iterations. The correlations were transformed using Fisher's *r* to *z* transformation, averaged, and then transformed back to *r*, yielding an average test-retest correlation whose maximum is limited by within-subject error.

## Experiment 2

Proportion correct, $p(c)$, in the complex AM detection results was transformed to $d'$ using the following equation from *Macmillan and Creelman, 2005*, pg. 172:

$$d' = \sqrt{2}z[p(c)]$$

Analyses were conducted using repeated-measures ANOVAs with type III sums of squares. Greenhouse-Geisser correction was used when Mauchly's test of sphericity was violated. Significant interactions were interpreted using post-hoc simple effects tests, with *p* values corrected using Holm's method (*Holm, 1979*).

## Acknowledgements

We thank Kara Stevens and Angela Sim for assistance with collecting data for Experiment 2 and Brian CJ Moore for thoughtful feedback on an early version of this manuscript. This work was supported by Grant R01 DC005216 from the National Institutes of Health (to AJO) and an Eva O Miller Fellowship (to KLW).

## Additional information

### Funding

| Funder | Grant reference number | Author |
|---|---|---|
| National Institutes of Health | R01 DC005216 | Andrew J Oxenham |
| University of Minnesota | Eva O. Miller Fellowship | Kelly L Whiteford |

The funders had no role in study design, data collection and interpretation, or the decision to submit the work for publication.

### Author contributions

Kelly L Whiteford, Conceptualization, Data curation, Software, Formal analysis, Methodology, Writing - original draft, Project administration, Writing - review and editing; Heather A Kreft, Conceptualization, Methodology, Writing - original draft, Project administration, Writing - review and editing; Andrew J Oxenham, Conceptualization, Resources, Supervision, Funding acquisition, Methodology, Writing - original draft, Writing - review and editing

### Author ORCIDs

Kelly L Whiteford https://orcid.org/0000-0002-2627-1509
Heather A Kreft https://orcid.org/0000-0003-0764-0820
Andrew J Oxenham https://orcid.org/0000-0002-9365-1157

### Ethics

Human subjects: Informed consent, and consent to publish, was obtained from all participants. All protocols were approved by the Institutional Review Board of the University of Minnesota (0605S85872).

### Decision letter and Author response

Decision letter https://doi.org/10.7554/eLife.58468.sa1
Author response https://doi.org/10.7554/eLife.58468.sa2

## Additional files

### Supplementary files

• Supplementary file 1. Table legends. (a) AM incoherence discrimination results. (b) Complex AM detection results. (c) Results with outlier subject included (n = 56) generally demonstrate the same trends as the main text, with the exception that the correlation between slow FM and slow AM detection thresholds was significantly different from the correlation between fast FM and fast AM detection thresholds (Z = 2.13, p=0.032, two-tailed), an effect that was not present with the outlier removed (see 'Correlations between FM and AM detection' in main text). (d) AM discrimination conditions. The units for all columns except Octave Separation are in Hz.

• Transparent reporting form

### Data availability

Source data files have been provided for Figures 2–6 and all figure supplements.

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
