## [Decision Letter]

**Acceptance summary:**

The inner ear can encode properties of a sound both through a place code as well as through a temporal code. In this article the authors show that, in contrast to the previous assumption, frequency fluctuations are detected primarily through place coding rather than temporal coding. This finding is particularly important since frequency modulations appear in many real-world sounds such as speech.

**Decision letter after peer review:**

Thank you for submitting your article "Perception of frequency modulation is mediated by cochlear place coding" for consideration by *eLife*. Your article has been reviewed by three peer reviewers, including Tobias Reichenbach as the Reviewing Editor and Reviewer #1, and the evaluation has been overseen by Andrew King as the Senior Editor. The following individual involved in review of your submission has agreed to reveal their identity: Enrique A Lopez-Poveda (Reviewer #4).

The reviewers have discussed the reviews with one another and the Reviewing Editor has drafted this decision to help you prepare a revised submission.

Summary:

The paper reports two experiments that support the hypothesis that a place code rather than a temporal code underlies the human ability to detect sound frequency modulations. The first experiment assesses people with normal hearing as well as participants with different degrees of hearing impairment, resulting in a wide range of fidelity of cochlear place coding. The authors find that the ability to detect frequency modulation is correlated to the fidelity of the cochlear place coding, even for low modulation frequencies where it was previously assumed that the place code did not matter. They conclude that cochlear place coding is critical for the detection of frequency modulation, at all modulation frequencies. In the second experiment, the authors show that a task of detecting amplitude modulation of two nearby frequencies with their amplitude modulation being either in our out of phase produces behaviour that is similar to that of detecting frequency modulation. Because the detection of amplitude modulation is not assumed to involve a temporal code, this is taken as evidence that the dependencies of the frequency modulation detection on the carrier frequency as well as on the modulation frequency does not need to result from temporal coding.

These results are overall well described and presented. The research question of place coding versus temporal coding regarding different aspects of a sound is important, since frequency modulation appears in many real-world sounds such as speech. Clarifying how these modulations are encoded in the brain can therefore help to elucidate the processing of many complex signals. However, there are a range of major issues that we would like the authors to address in a revised version.

1) The authors investigate detection thresholds for fast and slow FM as well as for fast and slow AM. The dual coding model predicts that the thresholds for detecting slow FM should not be correlated to the remaining three thresholds, whereas all other detection thresholds should be correlated to each other. In the author's model, all thresholds should be correlated to each other (including the thresholds for detecting slow FM). The authors find that the detection thresholds for slow FM and fast FM are significantly correlated, which corroborates their hypothesis. However, they also find that the correlation between the detection thresholds for slow FM and slow AM is insignificant, which seems to contradict their hypothesis. Moreover, the authors do not describe the correlation between the detection thresholds for slow FM and fast AM. Please detail this correlation and discuss how the seemingly diverging observations can be reconciled.

Hearing impairment may cause damage to the cochlear place coding, but also to temporal coding as well as to more central cognitive processes. Please discuss these possible confounds further.

2) The correlations presented in Figure 5 between the residuals of the FM detection and the slopes related to cochlear tuning appear to rely mostly on data points that have slopes around 0, or even of the opposite sign as the expected one. We are wondering what such slopes signify, and how they could be explained? It seems to us that they might indicate a more complicated pattern of hearing loss. Do we know – and can therefore control – how such more complicated hearing loss might affect FM or AM detection? Vice versa, would the correlations still persist if only slopes in a “reasonable” range, comparable to that of normal-hearing people, were to be included?

3) The second line of evidence uses multiple linear regression to account for the influence of other factors. This analysis produced one finding that is more in line with the traditional view: after hearing loss had been entered into the regression, sensitivity to AM accounted for 18.7% of the variance in fast-rate FM thresholds but only 4% of that in slow-rate FM thresholds. Could the authors compute whether, in the raw data, the correlation between the low-frequency slope and the FM thresholds was larger than that between the low-frequency slope and the AM thresholds? Such a correlation could be a clear prediction of the authors' hypothesis and might be quite powerful as it predicts that one across-listener correlation is significantly larger than another.

4) Regarding the second experiment, could combination tones have affected the performance? In addition, two important features of the design are missing from the main text and should be inserted: the use of exclusively normal-hearing subjects and the presence of a narrowband of noise between the two carriers.

5) We believe that the data may be insufficient to completely rule out a temporal code for FM in some conditions. Johannesen et al., 2016, which is not cited in the manuscript, reported that low-rate FM detection (*f_c_*=1.5 kHz, *f_m_*=2Hz) was not correlated with pure tone thresholds or age for hearing-impaired listeners (N=68), and was only slightly correlated with cochlear mechanical gain loss inferred using forward masking (N=68, R2=0.06, p=0.04, see their Table I). At first sight, the results of Johannesen et al., 2016, seem inconsistent with the data of experiment 1 in the present study. A potentially important difference between the two studies, however, is that Johannesen et al., 2016, superimposed random AM (rate 1-3 Hz, depth=6 dB) on their stimuli to minimize FM-to-AM conversion as a reliable cue for FM detection; i.e., to maximize the possibility that FM detection was based on an temporal code. Their hearing-impaired subjects were still able to perform the task. Therefore, the study of Johannesen et al. shows that a temporal code is likely used when FM-to-AM is not possible or reliable and makes us wonder if the pattern of results in experiment 1 would still hold if random AM were super-imposed on the stimuli.

The authors themselves acknowledge that FM is encoded in the timing of spikes of auditory neurons and indeed cite some studies that support a dual code for FM (e.g., Paraouty et al., 2018). Therefore, the more parsimonious explanation for all the evidence combined is that FM is probably encoded via a dual place-time code. In our opinion, the authors should transmit this idea more clearly rather than the idea that low-rate FM is encoded only via a place code.

[Editors' note: further revisions were suggested prior to acceptance, as described below.]

Thank you for submitting your article "The role of cochlear place coding in the perception of frequency modulation" for consideration by *eLife*. Your article has been reviewed by three peer reviewers, including Tobias Reichenbach as the Reviewing Editor and Reviewer #1, and the evaluation has been overseen by Andrew King as the Senior Editor. The following individual involved in review of your submission has agreed to reveal their identity: Enrique A Lopez-Poveda (Reviewer #4).

The reviewers have discussed the reviews with one another and the Reviewing Editor has drafted this decision to help you prepare a revised submission.

The reviewers agree that the revised manuscript is significantly improved. However, the reviewers still found the interpretation of the correlations between AM and FM detection at slow and fast rates too strong.

Unlike the correlations with spatial selectivity, overall effects of hearing loss are not partialled out, and so as a default (assuming that people with hearing loss have worse tuning and worse phase locking than those without) one might expect everything to correlate with everything else. Unfortunately, this is identical to the authors' hypothesis of what one would observe if FM and AM at slow and high rates were all processed using a unitary mechanism. In fact, even the prediction of everything correlating with everything else did not occur, and it seems that the results fit equally well or badly with both the authors' and the dual-mechanism explanations.

We previously pointed out one finding – the lack of correlation between slow AM and slow FM – that was inconsistent with the authors' interpretation (according to which everything should correlate with everything else). They now provide information that slow FM does not correlate with fast AM, also inconsistent with their interpretation. In their rebuttal they argue that the correlation between slow AM and fast AM was not significantly smaller than that between fast AM and fast FM. However, we note that a) it was significantly smaller when the "outlier subject" was not removed from the analysis, as shown in the Supplementary Information but not in the main text, and b) it was significantly smaller than some other correlations (e.g. slow FM vs fast FM).

This part of the argument therefore appears somewhat weak. It might be best just to describe the correlations, point out that one of them (slow vs fast FM) is not predicted by the dual-mechanism hypothesis, but note that the overall pattern does not strongly support one hypothesis or the other.

---

## [Author Response]

Revisions for this paper:1) The authors investigate detection thresholds for fast and slow FM as well as for fast and slow AM. The dual coding model predicts that the thresholds for detecting slow FM should not be correlated to the remaining three thresholds, whereas all other detection thresholds should be correlated to each other. In the author's model, all thresholds should be correlated to each other (including the thresholds for detecting slow FM). The authors find that the detection thresholds for slow FM and fast FM are significantly correlated, which corroborates their hypothesis. However, they also find that the correlation between the detection thresholds for slow FM and slow AM is insignificant, which seems to contradict their hypothesis. Moreover, the authors do not describe the correlation between the detection thresholds for slow FM and fast AM. Please detail this correlation and discuss how the seemingly diverging observations can be reconciled.Hearing impairment may cause damage to the cochlear place coding, but also to temporal coding as well as to more central cognitive processes. Please discuss these possible confounds further.

The correlations between slow FM and fast AM as well as fast FM and slow AM have been added to the experiment 1 results. The smaller correlations between slow FM and AM could be accounted for by a dual place-time code. Because the correlation between slow FM and slow AM was not significantly different from the correlation between fast FM and fast AM, however, the evidence for a dual code is modest, particularly in the light of the results from experiment 2. The trends with rate and center frequency in experiment 2 demonstrate that spike-driven timing is not necessary to explain rate and carrier dependent trends in FM sensitivity that have been typically attributed to time coding. The Results section on frequency selectivity in experiment 1 as well as the Discussion have been edited to point out that hearing impairment may co-vary with issues related to temporal coding and/or central processing. The title has also been changed to tone down the claim that only place coding can account for FM sensitivity. Importantly, even after controlling for potentially confounding effects of hearing loss, aging, and sensitivity to AM, the partial correlations between FM sensitivity and low-frequency masking pattern slopes persisted. This finding helps to establish that masking-pattern slopes affect FM sensitivity, even when the potential confounds of hearing loss, task demands of AM processing, and central effects of aging are accounted for.

2) The correlations presented in Figure 5 between the residuals of the FM detection and the slopes related to cochlear tuning appear to rely mostly on data points that have slopes around 0, or even of the opposite sign as the expected one. We are wondering what such slopes signify, and how they could be explained? It seems to us that they might indicate a more complicated pattern of hearing loss. Do we know – and can therefore control – how such more complicated hearing loss might affect FM or AM detection? Vice versa, would the correlations still persist if only slopes in a “reasonable” range, comparable to that of normal-hearing people, were to be included?

Because Figure 5 denotes the residuals (i.e., the difference between each individual data point and the predicted values) after controlling for sensitivity to AM at the same rate, age, and absolute thresholds for the carrier, the units here do not correspond directly to the slopes.

Figure 4, however, does show slopes, and a few subjects have slopes that are in the opposite of the expected direction, presumably due to measurement noise. Imputing these slopes with 0 (Author response image 1) or removing them entirely (Author response image 2) does not change the conclusions; this is now stated in the manuscript (subsection “Relationship between frequency selectivity and FM detection thresholds”). Restricting the data to only include those with normal hearing at the carrier would leave us with a substantially underpowered sample size, as most participants have hearing loss. However, past studies with only normal-hearing participants have shown no correlation between the steepness of the masking pattern slopes and fast or slow FMDLs (Whiteford and Oxenham, 2015; Whiteford et al., 2017), likely because there is not as much variability in peripheral place coding fidelity when all the listeners have healthy hearing.

**Author response image 1. sa2fig1:** Correlations between FM sensitivity (y-axes) and fidelity of place coding (x-axes) (n=55) with extreme slope data points (i.e., negative low slopes and/or positive high slopes) replaced with 0.

**Author response image 2. sa2fig2:** Correlations between FM sensitivity (y-axes) and fidelity of place coding (x-axes) with extreme slope data points removed.

3) The second line of evidence uses multiple linear regression to account for the influence of other factors. This analysis produced one finding that is more in line with the traditional view: after hearing loss had been entered into the regression, sensitivity to AM accounted for 18.7% of the variance in fast-rate FM thresholds but only 4% of that in slow-rate FM thresholds. Could the authors compute whether, in the raw data, the correlation between the low-frequency slope and the FM thresholds was larger than that between the low-frequency slope and the AM thresholds? Such a correlation could be a clear prediction of the authors' hypothesis and might be quite powerful as it predicts that one across-listener correlation is significantly larger than another.

The smaller amount of variance accounted for by slow-rate AM reflects the lower raw correlation shown in Figure 3. The correlations between the low-frequency slope and FM sensitivity was significantly stronger than that between the low-frequency slope and AM sensitivity for both slow (*Z* = -4.42; *p* <.0001) and fast rates (*Z* = -4.89; *p* <.0001), supporting the place coding view that FM sensitivity at both rates relies on frequency selectivity. This is now included in the Results section, “Relationship between frequency selectivity and FM detection thresholds.”

4) Regarding the second experiment, could combination tones have affected the performance? In addition, two important features of the design are missing from the main text and should be inserted: the use of exclusively normal-hearing subjects and the presence of a narrowband of noise between the two carriers.

The level of combination tones are dependent on both the ratio of the stimulus frequencies (*f1* and *f2*) and the stimulus level (e.g., Humes, 1989; Johnson et al., 2006) and tend to be maximal when the frequency ratio (f2/f1) is around 1.2 and when the level of f2 is about 20 dB higher than the level of f1. In our case, both tones were low in level (45 dB SPL). Even the smaller ratio of 2/3 octaves (f2/f1 = 1.59) is much greater than the ratios that generate combination tones, and the levels are too low. For the larger ratio of 4/3 octaves (f2/f1 = 2.52), no combination tones are mathematically possible, as the primary combination tone frequency (2f1-f2) falls below 0 Hz. In addition, the narrowband noise between the tones would further limit any potential combination tones.

The main text has been edited to state that experiment 2 used exclusively normal-hearing listeners and that narrowband noise was presented between the carriers to prevent any interactions between them, including combination tones.

5) We believe that the data may be insufficient to completely rule out a temporal code for FM in some conditions. Johannesen et al., 2016, which is not cited in the manuscript, reported that low-rate FM detection (f_c_=1.5 kHz, f_m_=2Hz) was not correlated with pure tone thresholds or age for hearing-impaired listeners (N=68), and was only slightly correlated with cochlear mechanical gain loss inferred using forward masking (N=68, R2=0.06, p=0.04, see their Table I). At first sight, the results of Johannesen et al., 2016, seem inconsistent with the data of experiment 1 in the present study. A potentially important difference between the two studies, however, is that Johannesen et al., 2016, superimposed random AM (rate 1-3 Hz, depth=6 dB) on their stimuli to minimize FM-to-AM conversion as a reliable cue for FM detection; i.e., to maximize the possibility that FM detection was based on an temporal code. Their hearing-impaired subjects were still able to perform the task. Therefore, the study of Johannesen et al. shows that a temporal code is likely used when FM-to-AM is not possible or reliable and makes us wonder if the pattern of results in experiment 1 would still hold if random AM were super-imposed on the stimuli.The authors themselves acknowledge that FM is encoded in the timing of spikes of auditory neurons and indeed cite some studies that support a dual code for FM (e.g., Paraouty et al., 2018). Therefore, the more parsimonious explanation for all the evidence combined is that FM is probably encoded via a dual place-time code. In our opinion, the authors should transmit this idea more clearly rather than the idea that low-rate FM is encoded only via a place code.

We agree with the reviewers that the superimposed random AM on FM makes it hard to directly compare Johannesen et al., 2016, to the present study. One issue with measuring FM with AM imposed is that it elevates FM thresholds at all carriers and rates relative to traditional FM thresholds – an effect that could be driven by more central aspects of FM processing, rather than less reliable place cues. Furthermore, superimposed AM on FM would not entirely wipe out FM-to-AM conversion; in fact, an optimal detector strategy could eliminate most of the AM interference by subtracting the coherent AM found on either side of the carrier frequency from the incoherent AM produced by the FM. Thus, any interference produced by AM may be the result of non-optimal processing, or (as suggested by our data) more reliance on the lower excitation pattern slope than the upper. In either case, the results from Johannesen et al. could potentially still be explained by the use of residual place cues and out-of-phase AM processing. The lack of correlation between pure-tone thresholds and FM thresholds in their study may be because their pure-tone thresholds did not specifically reflect thresholds at the carrier (i.e., they were weighted and averaged across multiple frequencies). The lack of correlation between FM thresholds and age is less clear and not consistent with what we (Whiteford et al., 2017) and others (Paraouty and Lorenzi, 2017) have found for traditional FM and age, although again this is made more difficult to interpret in that there are likely added demands on central processing when detecting FM with superimposed AM.

We have incorporated the Johannesen et al., 2016, paper into the manuscript and edited the Discussion to include the possibility that slow-rate, low-carrier FM uses a dual code that relies on combining both place and timing information, which may account for the patterns of correlations observed in experiment 1. Importantly, such a dual code could not account for the results of experiment 2, where no useful timing cues are available, calling into question the necessity of any timing information needed for slow-rate FM.

[Editors' note: further revisions were suggested prior to acceptance, as described below.]

Revisions for this paper:The reviewers agree that the revised manuscript is significantly improved. However, the reviewers still found the interpretation of the correlations between AM and FM detection at slow and fast rates too strong.Unlike the correlations with spatial selectivity, overall effects of hearing loss are not partialled out, and so as a default (assuming that people with hearing loss have worse tuning and worse phase locking than those without) one might expect everything to correlate with everything else. Unfortunately, this is identical to the authors' hypothesis of what one would observe if FM and AM at slow and high rates were all processed using a unitary mechanism. In fact, even the prediction of everything correlating with everything else did not occur, and it seems that the results fit equally well or badly with both the authors' and the dual-mechanism explanations.We previously pointed out one finding – the lack of correlation between slow AM and slow FM – that was inconsistent with the authors' interpretation (according to which everything should correlate with everything else). They now provide information that slow FM does not correlate with fast AM, also inconsistent with their interpretation. In their rebuttal they argue that the correlation between slow AM and fast AM was not significantly smaller than that between fast AM and fast FM. However, we note that a) it was significantly smaller when the "outlier subject" was not removed from the analysis, as shown in the Supplementary Information but not in the main text, and b) it was significantly smaller than some other correlations (e.g. slow FM vs fast FM).This part of the argument therefore appears somewhat weak. It might be best just to describe the correlations, point out that one of them (slow vs fast FM) is not predicted by the dual-mechanism hypothesis, but note that the overall pattern does not strongly support one hypothesis or the other.

We have tempered the interpretation of the correlations between AM and FM detection at slow and fast rates in the Results and the Discussion.